# The immune response to RNA suppresses nucleic acid synthesis by limiting ribose 5-phosphate

Pushpak Bhattacharjee[1], Die Wang [1], Dovile Anderson[2], Joshua N Buckler [3], Eveline de Geus [1], Feng Yan [4], Galina Polekhina[5], Ralf Schittenhelm[6], Darren J Creek[2], Lawrence D Harris [3] & Anthony J Sadler [1]✉

## Abstract

**During infection viruses hijack host cell metabolism to promote their replication. Here, analysis of metabolite alterations in macrophages exposed to poly I:C recognises that the antiviral effector Protein Kinase RNA-activated (PKR) suppresses glucose breakdown within the pentose phosphate pathway (PPP). This pathway runs parallel to central glycolysis and is critical to producing NADPH and pentose precursors for nucleotides. Changes in metabolite levels between wild-type and PKR-ablated macrophages show that PKR controls the generation of ribose 5-phosphate, in a manner distinct from its established function in gene expression but dependent on its kinase activity. PKR phosphorylates and inhibits the Ribose 5-Phosphate Isomerase A (RPIA), thereby preventing interconversion of ribulose- to ribose 5-phosphate. This activity preserves redox control but decreases production of ribose 5-phosphate for nucleotide biosynthesis. Accordingly, the PKR-mediated immune response to RNA suppresses nucleic acid production. In line, pharmacological targeting of the PPP during infection decreases the replication of the Herpes simplex virus. These results identify an immune response-mediated control of host cell metabolism and suggest targeting the RPIA as a potential innovative antiviral treatment.**

Keywords Antiviral; Immunity; Metabolism; Nucleotide Biosynthesis; PPP (Pentose Phosphate Pathway); PKR (Protein Kinase RNA-Activated); RPIA (Ribose 5-Phosphate Isomerase A)
**Subject Categories** Immunology; Metabolism; Microbiology, Virology & Host Pathogen Interaction

## Introduction

Glucose utilization is controlled through conserved cell signaling pathways that converge on crucial metabolic nodes. Protein translation initiation is a key node, likely, because protein synthesis constitutes a major metabolic cost to the cell. This checkpoint is controlled through the opposing action of the mechanistic target of rapamycin in complex 1 (mTORC1) and the kinases of the eukaryotic initiation factor 2α (EIF2α, also abbreviated EIF2S1). The metabolic impact of this dyad has largely focused on mTORC1 with comparatively less attention on the effects of the EIF2α kinases.

The primal EIF2α kinase General Control Nonderepressible 2 (GCN2), sensing low amino acid levels, induces a starvation response by inhibiting the reformation of the 43S preinitiation complex through phosphorylation of EIF2α (Chen et al, 1991; Roussou et al, 1988). Multiple other EIF2α kinases have arisen through gene duplications that have then been fixed in the genome by the acquisition of different sensor domains that broaden the starvation response, which is controlled by GCN2 (Muaddi et al, 2010). Three additional EIF2α kinases occur in mammals. The Heme-Regulated Inhibitor, Protein Kinase RNA-activated (PKR), and PKR-like Endoplasmic ReticulumKinase responds to the levels of heme, duplex RNA, or unfolded proteins, respectively (Berlanga et al, 1998; Metz and Esteban, 1972; Shi et al, 1998). The sensing of duplex RNA by PKR equips it to detect viral replicative intermediates (McCormack et al, 1992). This activity is reinforced by the induction of PKR by the antiviral type I and III interferons (Roberts et al, 1976), making the kinase an important part of our antiviral response. Despite the overt consequences, there has been little consideration of the metabolic impact of PKR during infection.

Here we recognize that PKR alters glucose metabolism. Upon uptake, glucose is primarily oxidized to a three-carbon compound via central glycolysis to provide adenosine triphosphate (ATP) and metabolic intermediates or is used in the pentose phosphate pathway (PPP) to yield ribose sugars and reducing agents for biosynthetic processes and redox control. During infection, viruses

[1]Centre for Innate Immunity and Infectious Diseases, Hudson Institute of Medical Research and Department of Molecular and Translational Sciences, Monash University, Clayton, VIC 3168, Australia. [2]Drug Delivery, Disposition and Dynamics, Monash Institute of Pharmaceutical Sciences, Monash University, Parkville, VIC 3052, Australia. [3]Ferrier Research Institute, Victoria University of Wellington, Lower Hutt 5010, New Zealand. [4]Australian Centre for Blood Diseases, Department of Clinical Hematology, Monash University, Clayton, VIC 3004, Australia. [5]Department of Epidemiology & Preventive Medicine, Monash University, Melbourne, VIC 3004, Australia. [6]Monash Proteomics & Metabolomics Facility, Department of Biochemistry and Molecular Biology, Biomedicine Discovery Institute, Monash University, Clayton, VIC 3800, Australia. ✉E-mail: anthony.sadler@hudson.org.au; anthonysadler630@gmail.com

hijack the host cell metabolism to promote their replication by altering the breakdown of glucose (Abrantes et al, 2012; Landini, 1984; Vastag et al, 2011; Xie et al, 2017), with metabolic analysis identifying anabolic processes are boosted through increased PPP activity (Chen et al, 2011; Delgado et al, 2012; Guo et al, 2019; Liu et al, 2015; Shi et al, 2016; Wang et al, 2016b; Yau et al, 2021). We identify that PKR counters this by selectively limiting a pentose precursor molecule that is required by viruses to replicate their genome. The findings identify a previously unknown immune-mediated metabolic adaptation to limit the replicative capacity of viruses during infection that has the potential to treat infectious pathogenesis. This would meet a clear unmet medical need for a broad-spectrum treatment against emergent viral diseases.

## Methods

### Reagents

Clones of human and murine genes were purchased as carboxyl-terminal, FLAG-tagged constructs (OHu27509, OHu00691, OHu13849, OHu19612, OHu18642D, OHu02133D, OHu29487D, OHu03814, and OHu62291 from GenScript and MR207793, MR225480, MR204498, MR227514, MR217037, MR204296, and MR208390 from OriGene). *E. coli* genes were acquired from the National BioResource Project, National Institute of Genetics (Japan) (Appendix Table S1). Endogenous transcripts were targeted by RNA interference using pre-designed lentiviral constructs (clone ID TRCN0000026988, TRCN0000274652, or TRCN0000274653 targeting *Eif2ak2*, TRCN0000301646, TRCN0000301721, or TRCN0000071724 targeting *Atf4* and SHC016-1EA as a non-targeting control, Sigma-Aldrich). Plasmids were isolated in chemically competent *E. coli* TOP10 (Thermo Fisher Scientific), JM109 (Promega), or Stbl3 (Thermo Fisher Scientific), with the latter used for lentiviral and shRNA constructs. Restriction enzymes, T4 DNA ligase, and antibiotics were purchased from Bio-Rad. Carbohydrates and other chemicals were purchased from Sigma-Aldrich unless otherwise stated. D-Sedoheptulose was purchased from Biosynth Carbosynth Ltd. Restriction enzymes were purchased from New England Biolabs. We use Eppendorf brand polypropylene tubes and tips.

### Bacterial assays

Human PKR was expressed in the *E. coli* strain BL21(DE3)-pLysS for growth and complementation assays, although experiments measuring the growth on a single carbon source used a derivative strain, C41(DE3), to escape a disabling transposon in the ribose operon of BL21 (Cooper et al, 2001). *EIF2AK2* and *RPIA* were expressed in *pET15b* by induction with IPTG (0.1 or 0.3 mM for BL21 or C41, respectively, Bio-Rad). *EIF2AK2* was co-expressed with the bacteriophage λ phosphatase as described previously (Matsui et al, 2001), although the genes were subcloned into *pET15b*. PPP genes were cloned and expressed in the *pWKS130* vector as *Kpn*I/*Sal*I-*Eco*RI fragments (Wang and Kushner, 1991). Notably, the respective bacteriophage *f1* and *pMB1* replication origins that are used in each plasmid can be simultaneously maintained. Clones were verified by DNA sequencing at the Micromon Genomics facility at Monash University. To aid

template preparation for sequencing, we increase the copy number of *pWKS130* (from ~27 to ~84 copies/cell) by replacing glutamate with glycine at position 93 in the replication initiator protein (Peterson and Phillips, 2008). Bacterial transformants were isolated on restrictive Luria-Bertani (LB) agar plates containing chloramphenicol (34 μg/mL for the T7 lysozyme), carbenicillin (100 μg/mL for *pET15b*) and/or kanamycin (50 μg/mL for *pWSK130*). The following day a single colony was re-streaked and regrown on restrictive LB agar plates. After overnight growth, a single bacterial colony was inoculated into liquid M9 medium (42.2 mM $Na_2HPO_4$-$7H_2O$, 147 mM $KH_2PO_4$, 8.6 mM NaCl, 18.7 mM $NH_4Cl$, 0.1 mM $CaCl_2$, 1 mM $MgSO_4$, 3.8 μM thiamine, 1 μg/mL, 35–70 μM carbon source) supplemented with antibiotics with shaking at 37 °C. The following day the culture was diluted 1 in 400 and regrown in M9 medium with antibiotics, with and without IPTG. Bacterial growth was measured by the optical density at 600 nm.

*E. coli* BL21(DE3)-pLysS-*pET15b-EIF2AK2* and strains containing conjugal plasmids encoding metabolic factors cloned into the vector *pNTR-SD* or *pCA24N* (Appendix Table S1 (Saka et al, 2005)) were cultured overnight in liquid LB medium supplemented with antibiotics at 37 °C with shaking. The following day the cultures were diluted 1 in 50 and cultured in LB until visibly turbid (OD600 = 0.013), then the donor and recipient strains were combined 1 to 1 or diluted 1 in 2 with LB medium and cultured for a further hour at 30 °C. The three different cultures were diluted 1 in 10, then 100 μL was spread onto LB agar plates supplemented with carbenicillin (100 μg/mL selecting for each plasmid) or chloramphenicol (50 μg/mL, restrictive for the BL21(DE3)-pLysS strain) or these two antibiotics combined, with and without IPTG (0.5 mM). Plates were cultured overnight at 37 °C, and the growth of transconjugants with antibiotics and IPTG was attributed to relief of PKR-dependent repression by the specific mobile genetic element on the conjugal plasmid. Verification of the acquisition of plasmids by BL21(DE3)-pLysS expressing PKR is shown as supplementary data (Appendix Fig. S3). Three independent experiments were conducted, and representative results are shown.

### Viral infection

MEFs were seeded into culture plates with Dulbecco's modified Eagle's Medium (DMEM, GibCo) supplemented with 10% fetal bovine serum (Bovogen) in a humidified 5% $CO_2$ incubator at 37 °C for 24 h, at which point the cells were 80% confluent. The cells were then treated with either the solvent ($H_2O$) or the prodrug (200 μM) or an approximate clinical dose of the recombinant antiviral cytokine IFNβ (100 iu/mL), followed 60 min later by infected with GFP-HSV-1 at an MOI of 0.1 (Elliott and O'Hare, 1999). Infected cells were monitored for the development of GFP fluorescence over 48 h, at which point all cells expressed the viral reporter, and virus production was quantified by endpoint titration of virions in the culture medium against Vero cells by established protocols (Witek and Wieczorek, 2009). Infection of Vero cells was evidenced as fluorescence after a week of culture. Fluorescent signals were recorded using a Zeiss MBQ 52 AC burner on an Axiovert 40 CFL Trinocular Inverted fluorescence microscope and a ProgRes camera and software (Jenoptik). The data were analyzed using the ImageJ software (NIH and LOCI).

## Protein preparation

Recombinant proteins for enzyme assays were expressed and purified in *E. coli* BL21(DES)-pLysS. Human PKR and Protein phosphatase 2 regulatory subunit B alpha (PPP2R5A or B56α) were expressed as a hexa-HIS-tagged protein and recovered with Ni-NTA HIS-Bind resin (Merck) as previously described (Xu et al, 2004). Human RPIA was expressed as a Streptavidin-tagged protein in a modified *pET15b* vector and extracted with IBA Strep-Tactin XT 4flow resin according to the manufacturer's protocol (IBM Lifesciences). FLAG-tagged proteins were prepared from HEK293 cells for in vitro isomerase and kinase assays. The different constructs in the *pcDNA3.1* vector were transfected into $1 \times 10^6$ cells with lipofectamine 2000 (Thermo Fisher Scientific), then after 48 h, the cells were lysed with RIPA buffer (50 mM Tris-HCl, pH 7.4, 150 mM NaCl, 0.25% deoxycholic acid, 1% NP-40, 1 mM EDTA) supplemented with the Halt™ protease (Life Tech). Cell debris was removed by centrifugation, and lysates were precleared with protein A/G–agarose beads (Pharmacia Biotech) then tagged proteins were recovered with FLAG-conjugated agarose beads (Sigma-Aldrich) at 4 °C for 6 h. After five washes with PBS, proteins were resuspended in 10 mM Tris-HCl (pH 7.6).

## Enzyme assays

Isomerase enzymes were enriched from transfected HEK293 cells (expression from *pcDNA3.1*) or transduced *E. coli* BL21(DES)-pLysS (expression from *pET15b*) as either FLAG- or Streptavidin-tagged proteins, respectively. The activity of the enriched proteins was assessed using an in vitro spectrophotometric assay following an established protocol (Knowles et al, 1969; Wood, 1970).

Levels of flavin adenine dinucleotide (FAD) were measured by oxidase assay by the quantification of a fluorescent (OxiRed) probe using a commercial assay kit (Abcam ab204710) following the manufacturer protocol.

Levels of the reduced and oxidized form of nicotinamide adenine dinucleotide phosphate (NADP+ and NADPH) were measured by an enzyme cycling reaction by bioluminescence using a commercial assay kit (Abcam ab176724) following the manufacturer protocol.

## Assays in mammalian cells

Three separate murine macrophage cell lines were used; immortalized lines isolated from the spleens of WT or *Eif2ak2*$^{-/-}$ mice (Chakrabarti et al, 2008; Yang et al, 1995) and a line established from the bone marrow of WT mice (De Nardo et al, 2018). Exogenous genes were stably expressed, or endogenous gene expression was suppressed by the transduction of macrophages with lentiviral constructs. Gene open reading frames were cloned as *Nhe*I-*Bam*HI (New England Biolabs) fragments in the *plenti-CRISPR v2-Blast* that were expressed as self-cleaving (P2A) blasticidin selectable polypeptides (Mohan Babu, Addgene plasmid # 83480, http://n2t.net/addgene:83480, RRID Addgene_83480). Knockdown of endogenous transcripts was achieved with short hairpin RNA cloned into *pLKO.1-puro*. Re-expression of a suppressed factor was achieved by targeting outside of the gene open reading frame, in the untranslated region, to avoid the expressed transcript. Lentiviral particles were packaged using a

Lenti-X mix (Takara) by transfection of the plasmid components into HEK293FT cells with lipofectamine 2000. The culture supernatant was collected after 48 h, centrifuged at 10,000×*g*, and then pipetted onto adherent macrophages. Transduced cells were isolated over 2–3 months by increasing selection (from 1 to 20 µg/mL) with puromycin (Thermo Fisher Scientific) and/or blasticidin (InvivoGen), for *pLKO-1-puro-shRNA* and *plentiCRISPR v2-blast*, respectively. Recombinant cells were maintained with antibiotics. Growth on different carbon sources was tested by plating $6 \times 10^3$ cells per well of a 24-well plate with glucose-free DMEM (Thermo Fisher Scientific) supplemented with 10% FBS (not dialyzed), 20 µg/mL puromycin and 20 µM of each sugar. After 72 h, the cells were lifted with trypsin and counted by automation (Countess, Invitrogen). PKR was transiently activated in the splenic macrophages by transfecting cells with polyinosinic-polycytidylic acid (poly I:C 2 µg/mL) complexed with lipofectamine 2000 or persistently activated by duplex RNA produced from transfected expression vectors in the bone marrow-derived macrophages or HEK293 cells (De-Souza et al, 2019; Nejepinska et al, 2014).

## Metabolic analysis

Comparative measures of metabolites in WT and *Eif2ak2*$^{-/-}$ macrophages were conducted at the Monash Proteomics and Metabolomics Facility at the Monash Institute of Pharmaceutical Sciences, Monash University. Adherent macrophages were lifted with trypsin, suspended in ice-cold PBS, and $2 \times 10^6$ cells were collected by centrifugation. The cell pellet was rinsed in 10 mL ice-cold 0.9% NaCl in water at 4 °C, pelleted, and then resuspended in 200 µL of 4 °C extraction solvent (a 1:3:1 ratio of chloroform, methanol, and water). The sample was subjected to three freeze-thaw cycles and then mixed by vortex for 30 min at 4 °C, and cell debris was removed by centrifugation at 20,000×*g* for 10 min at 4 °C. The cleared supernatant was transferred (180 µL) and frozen at −80 °C until LCMS analysis. Samples were prepared from four independent experiments with WT and *Eif2ak2*$^{-/-}$ cells processed in parallel to minimize procedural effects. A blank control, without cells but using the same reagents, was also prepared in parallel to measure background contaminants. LCMS acquisition used Dionex Ultimate 3000 RSLC liquid chromatography system and QExactive (Thermo Fisher Scientific) mass spectrometer, with SeQuant ZIC-pHILIC 5 µm 150 × 4.6 mm analytical column and ZIC-pHILIC guard column. Data quality is assessed by inspecting peak median within sample groups, reproducible QC pooled samples, and inspecting heatmap for outliers. The data were analyzed with MZmine (Pluskal et al, 2010) and IDEOM (Creek et al, 2012) with metabolite identification based on accurate mass and retention time relative to a library of authentic standards (level 1) or a database of predicted retention times (level 2) (Creek et al, 2012). This data is available at the NIH Common Fund's National Metabolomics Data Repository website, the Metabolomics Workbench, https://www.metabolomicsworkbench.org as project ID ST002412 and https://doi.org/10.21228/M8342Z.

## Phosphorylation analysis

Phosphorylation was assessed through radiolabeling by in vitro kinase assay, visualization by phosphoprotein stain, isoelectric focusing, and mass spectrometry analysis.

## Kinase assays

Kinase assays followed an established procedure (Hovanessian et al, 1983). Briefly, recombinant HIS-tagged PKR (100 ng) and each FLAG-tagged PPP enzyme or the established substrates EIF2α and PPP2R5A (or B56α) (200 ng) were combined in the kinase buffer (10 mM Tris-HCl [pH 7.6], 50 mM KCl, 2 mM Mg(OAc)$_2$, 7 mM 2-Mercaptoethanol, 20% Glycerol, 1.67 mM MnCl$_2$, 1 μM ATP, 10 μCi of [γ-$^{32}$P]ATP (6000 Ci/mM, PerkinElmer), with poly I:C (60 ng) for 15 min at 30 °C. The reaction was stopped by the addition of 500 μL of 4 °C PBS with 500 mM EDTA and FLAG-tagged proteins were immunoprecipitated with an anti-FLAG antibody. Proteins bound to FLAG-conjugated agarose beads were washed five times with PBS with EDTA before denaturing in Laemmle buffer at 95 °C for 5 min. Immune precipitates were separated by SDS-PAGE electrophoresis and the gel was wrapped in plastic and exposed to autoradiographic film to detect radiolabeled products.

## Phosphoprotein stain

Phosphorylation of RPIA in *E. coli* was assessed by recovering the FLAG-tagged protein from lysates of BL21(DE3)-pLysS co-expressing WT or kinase-dead PKR with an anti-FLAG antibody at room temperature. *E. coli* were lysed with BugBuster supplemented with Benzonase and a phosphatase inhibitor cocktail (Sigma-Aldrich). Immune precipitates were washed five times in PBS, then denatured and separated by SDS-PAGE electrophoresis and visualized with the Pro-Q Diamond phosphoprotein stain following the manufacturer's protocol (Thermo Fisher Scientific). Stained phosphoproteins were detected with a FujiFilm FLA-5100 fluorescent image analyzer (with filters: LPG/0575#50-000/01 and IP#17-000/01). The image was captured with Image Gauge v2.2 software (Fuji Pharma) and the figure was prepared with Pixelmator Pro Odesa (Pixelmator).

## Isoelectric focusing

Isoelectric focusing of proteins was performed using a ZOOM IPG Runner system following the manufacturer's protocol (Invitrogen). Briefly, cleared whole cell lysates from $5 \times 10^6$ WT or *Eif2ak2$^{-/-}$* spleen-derived macrophages treated with poly I:C (2 μg in lipofectamine reagent) for four hours were focused with the ZOOM IEF fractionator (pH 3–10), further separated by SDS-PAGE electrophoresis through a Novex 4–12% Tris-glycine protein gel, then transferred to membrane for immune detection of proteins with specific antibodies.

## Mass spectrometry

Mass spectrometric identification of phosphorylated residues was performed at the Monash Proteomics and Metabolomics Facility at Monash University. To isolate RPIA for analysis, $1 \times 10^6$ HEK293 cells with suppressed PKR expression (as previously reported (Liu et al, 2014)) were transfected with the complementary split Venus-tagged PKR and RPIA constructs (100 ng and 500 ng, respectively, in lipofectamine reagent) in a 5 cm dish. After 48 h, the cells were rinsed with PBS, then lysed in RIPA supplemented with a

phosphatase inhibitor cocktail and cleared by centrifugation. Protein complexes were captured by GFP-Trap (ChromoTek) and digested with trypsin (Promega) after disulfide bridges were reduced and alkylated with TCEP (Thermo Fisher Scientific) and CAA (Sigma), respectively. Using a Dionex UltiMate 3000 RSLCnano system equipped with a Dionex UltiMate 3000 RS autosampler, an Acclaim PepMap RSLC analytical column (75 μm × 50 cm, nanoViper, C18, 2 μm, 100 Å, Thermo Fisher Scientific) and an Acclaim PepMap 100 trap column (100 μm × 2 cm, nanoViper, C18, 5 μm, 100 Å, Thermo Fisher Scientific), the tryptic peptides were separated by increasing concentrations of 80% acetonitrile (ACN)/0.1% formic acid at a flow of 250 nL/min for 98 min and analyzed with a Orbitrap Fusion Tribrid mass spectrometer (Thermo Fisher Scientific) operated in data-dependent acquisition (DDA) mode. The mass spectrometric raw files were analyzed with Byonic (ProteinMetrics) using custom-made databases containing the split Venus-tagged RPIA and the WT or kinase-dead, point mutant (K296R) PKR sequences (Appendix Fig. S10). Fixed modifications were applied for cystine derivation (carbamidomethylation), and variable modifications were tested for oxidation of methionine, acetylation at the N-terminus, and phosphorylation of serine, threonine, and tyrosine residues. The false discovery rate was set to 1%. These data were available at the ProteomeXchange Consortium, https://www.proteomexchange.org via the PRIDE (Perez-Riverol et al, 2022) partner repository with the dataset identifier PXD036779.

## Split Venus assay

Split Venus constructs were made by cloning into the *Acc*III-*Xba*I or *Not*I-*Cla*I restriction endonuclease sites of *pcDNA3* containing the Venus residues 1–210 (V1) or residues 210–238 (V2) (Hu et al, 2002). Human PKR was tagged at its amino terminus with V2. Human RPIA was tagged at either terminus with V1, with no difference apparent between the amino- and carboxyl-tagged constructs. The *E. coli* RpiA was tagged at its amino terminus. Colocalization of the split Venus-tagged proteins was visualized by transfecting $1 \times 10^5$ HEK293 seeded onto sterile coverslips (Menzel-Glaser, 12 mm diameter, #1.5, Thermo Fisher Scientific) within a 24-well culture plate with 20 ng/well of the PKR construct and 50 ng/well of the RPIA construct using lipofectamine 2000. After 48 h, cell nuclei were stained with Hoechst 33342 (Invitrogen) for 10 min, then the cells were rinsed with phosphate buffer and fixed in 4% paraformaldehyde for 20 min, rewashed with PBS and mounted in 13% Mowiol, 33% glycerol, and 20% sodium azide (pH 8.5). Fluorescent images were recorded with an Olympus U-RFL-T burner and a BX60 microscope using a DP74 camera and the Olympus CellSens software.

## Flow cytometry

Single-cell suspensions of macrophages ($5 \times 10^6$) were prepared by trypsin treatment, then washed in PBS by suspension then centrifugation, followed by staining with propidium iodide, SYTO™ RNASelect™ or Hoechst 33342 (Thermo Fisher Scientific). Data were acquired on a BD LSR Fortessa X-20 and analyzed by FlowJo™ Software (BD Biosciences) at the Monash Flowcore, Monash Health Translational Precinct, Translational Research Facility.

## Molecular modeling

To predict how PKR and RPIA may interact with each other, we used protein docking and molecular superposition (Hex 8.0.0 (Ritchie and Kemp, 2000)) with the structure of a PKR monomer with ATP analog from RCSB PDB entry 2A19 and a model of human RPIA produced by SWISS-MODEL server based on the *E. coli* RpiA (RCSB entry 1LKZ). In the docking procedure, the PKR monomer is kept fixed, and the full rotational search is performed on RPIA. All possible docking orientations of RPIA are then scored based on surface complementarity and electrostatic potential, and multiple solutions that are close in their orientation are clustered. The top solutions are then filtered and inspected to identify the RPIA residues for plausible phosphorylation based on the proximity to the active site of PKR and ATP. The model of the protein complex was generated using PyMOL (The PyMOL Molecular Graphics System, Version 1.2r3pre, Schrödinger, LLC). PDB files of the top four scored protein complexes are shown as Source Data 7.

## Bioinformatic comparisons of proteins

The amino acid sequences of the human, mouse and *E. coli* isomerases were aligned using T-Coffee (https://tcoffee.crg.eu/apps/tcoffee/all.html). In addition, the conservation of isomerases from diverse organisms was assessed by alignment of the ternary structure of the *E. coli* protein (1LKZ) against the predicted structure of the human enzyme by Foldseek (https://foldseek.com).

## Chemical synthesis

Reactions requiring anhydrous conditions were carried out in flame-dried glassware under a positive pressure of argon in anhydrous solvents, using standard Schlenk techniques. Reaction temperatures above room temperature were carried out in heating mantles with an internal temperature probe. Reaction progress was monitored by either thin layer chromatography (TLC) on Merck Aluminum-backed silica gel-coated TLC plates (60 Å, F254 indicator) or by LCMS. TLC plates were visualized by exposure to ultraviolet light (254 nm), and/or $KMnO_4$. LCMS analysis was performed on an Agilent 1260 Infinity II Series LC System with an Agilent 6120B Single Quadrupole LC/MS (ESI), equipped with an Agilent 1100 Multi-Wavelength Detector and an Agilent Infinity II 1290 Evaporative Light Scattering Detector using a C18 Kinetex column ($50 \times 3$ mm, 2.6 µm) with a linear gradient system (solvent A: 0.1% (v/v) formic acid in water, solvent B: MeOH, 5–100% B over 6 min) at a flow rate of $1 \, mL \, min^{-1}$. Flash column chromatography was performed with a Büchi Pure C-815 Flash automated flash chromatography system using prepacked Flash-Pure cartridges containing either silica gel (50 µm irregular) or C18 silica gel (50 µm spherical), using ACS grade solvents or by passing through a Waters Corp Sep-Pak C18 Plus Long Cartridge. All yields refer to chromatographically and spectroscopically ($^1$H and $^{13}$C($^1$H) NMR) pure material. NMR spectra were recorded using a Bruker 500 MHz spectrometer. All chemical shifts (δ) are reported in parts per million (ppm) and referenced to residual protium or the carbon resonance of the NMR solvent, respectively. $^{31}$P($^1$H) NMR spectra were referenced to $H_3PO_4$ as an external standard. Data were represented as follows: chemical shift, multiplicity (br=broad, s=singlet, d=doublet, t=triplet, q=quartet, m=multiplet), coupling constants (J) in Hertz (Hz), integration. High-resolution electrospray ionization (ESI) mass spectra were undertaken on a Waters Q-TOF Premier™ Tandem Mass spectrometer fitted with a Waters 2795 HPLC. Optical rotations were measured on a Rudolph Research Analytical Autopol IV automatic polarimeter. The phosphoramidate prodrug JNB-hu20-051A was synthesized similarly, utilizing the methodology developed previously (Ross et al, 2011). The JNB-hu20-057-2A phosphate was synthesized according to established protocols (Dardonville et al, 2004; Yep et al, 2011). Detailed protocols for chemical synthesis are described as supplementary material (Appendix Supplementary Methods).

## Western blotting

Denatured samples were separated by SDS-PAGE and transferred to Immobilon-FL Membrane (Millipore). Membranes were probed with primary antibodies; anti-PKR (sc-6282, Santa Cruz; EPR19374, Abcam; #12297 D7F7, Cell Signaling Technology or BC1 (Zamanian-Daryoush et al, 1999)), -G6PD (#12263 D5D2, Cell Signaling Technology), -PGLS (NBP2-19785, Novus Biologicals), -PGD (NBP1-31589, Novus Biologicals), -RPIA (ab229967, Abcam or HPA042620, Sigma), -RPE (ab128891, Abcam or PA5-57689, Thermo Fisher Scientific), -TALDO1 (ab187130, Abcam), -TKT (PA5-56166, Thermo Fisher Scientific or HPA029481, Merck), -EIF2α (AHO0802, Thermo Fisher Scientific), -EIF2B4 (PA5-71496, Thermo Fisher Scientific), -ATF4 (PA5-19521, Thermo Fisher Scientific or #11815 D4B8, Cell Signaling Technology), and -ATF3 (PA5-36244, Thermo Fisher Scientific), -FLAG (a2220, Merck). Primary antibody complexes were visualized with either LI-COR IRDye or StarBright fluorescent secondary antibodies diluted 1:10,000 and visualized and quantified with the Odyssey (LI-COR, USA) or ChemiDoc (Bio-Rad) imaging.

## Quantitative real-time PCR

Total RNA was isolated using the TRIzol reagent as per the manufacturer's protocol (Invitrogen). The sample was treated with a TURBO DNA-free Kit (Ambion) and cDNA was synthesized using the PrimeScript RT Reagent Kit (TaKaRa). Quantitative PCR (Q-PCR) was performed with SYBR Green (Applied Biosystems, USA) using an Applied Biosystems 7700 Prism real-time PCR machine at the Medical Genomics Facility within the Hudson Institute of Medical Research. The *18S* ribosomal subunit and *Glucose 6-Phosphate Isomerase 1* (*Gpi1*) transcripts were used as comparisons. Results are expressed as relative gene expression compared to the untreated WT cell as triplicates of each target. PCR primers used were; *G6pd*-F CCACAGTCTATGAAGCAGTCAC, *G6pd*-R CATCTCTTTGCCCAGGTAGT, *Pgd*-F GAGCGAAACCCAGAACTTCA OR- TGCTTCCAAGATCATCTCCTAC, *Pgd*-R GCATCTCGTGTCTGTACCCAT OR- TGAAGTTCTGGGTTTCGCTC, *Pgls*-F CCAGGTCCTTACCATCAATCC, *Pgls*-R CACGATCTTCTCCCGCT, *Rpe*-F GCACATGATGGTGTCTAGGC, *Rpe*-R CTGGTTTGATGGCAAGGC, *Rpia*-F GCATCCCCACATCTTTCCAG OR- GGTGGCACAAGGGGATTC, *Rpia*-R GCATAACCAGCCACGATCT OR- CACGCCTGGAGTCATTTGA, *Tkt*-F ATCATCGTGGACGGACACAG, *Tkt*-R CGCCTCCTTGTCTTCAATCC, *Taldo1*-F GCGAGATCAAAGCACTGG, *Taldo1*-R TCT

TCTCCGAGTCACTGGT, *Gpi1*-F GACCAGCACTTCCTCAAGA, *Gpi1*-R TGTACTTTCCGTTGGACTCC, *18S*-F GGCCTCGAAAG AGTCCTGTA, *18S*-R AAACGGCTACCACATCCAAG (Bioneer Pacific).

## Construction of mutant proteins

Amino acid substitutions and truncations were produced by site-directed amplification using the following primers with the complementary reverse sequence; RPIA-δ75-F TGACGCGGGATC CATGTCCAAGGCCGAGGAGGCC, FLAG-RPIA-R TTGCTGAA TTCTTATCACTTATCGTCGTCATCCTTGTAATC, RPIA-S147 A-F GCAGTATGGCTTGACCCTCGCTGATCTGGATCGACACC CA, RPIA-K173R-F GCTGATCTCAATCTCATCAGGGGTGGC GGAGGCTGC, RPIA-S302A-F ATGCAGGATGGCGCAGTGA ACATGAGG, RpiA-S75A-F GATCTCAACGAAGTCGACGCCC TTGGCATCTACGTTGATGG, RpiA-T210A-F GGACGTTGCG CTGATTGGCGCACCTGACGGTGTCAAAACC, TALDO1-S171 F-F GCAACATGACGTTACTCTTCTTCTTCGCCCAGGCTGTG G, TKT-D155A-F GTCTATTGCTTGCTGGGAGCCGGGGAGCT GTCAGAGG (Bioneer Pacific).

## Statistical analysis

The statistical difference between groups was performed by the two-tailed student's *t*-test. Differences between three or more groups were assessed by the analysis of covariance. Analysis of multiple data points temporally collected was performed by comparisons of the slopes of fitted trend lines (t-statistic). LCMS metabolic data were processed with the IDEOM data analysis platform. Data were analyzed with Microsoft Excel v16.26 Analysis ToolPak software. All experiments were performed independently a minimum of three times. Results are given as the $\bar{x} \pm \sigma$. $P$ values of less than 0.05 were considered significant, and the specific calculated value is given or, if <0.001, recorded as such in each figure. Graphs generated in Microsoft Excel were formulated as figures with Adobe Illustrator.

# Results

## PKR alters glucose metabolism

We analyzed the activity of PKR in macrophages as the metabolism of this immune cell type is stringently controlled (Covarrubias et al, 2016; Ecker et al, 2010; Newsholme et al, 1986). Immortalized macrophage lines isolated from the spleens of either wild-type (WT) or *Eif2ak2*$^{-/-}$ (the gene encoding PKR) mice were treated with the RNA mimetic poly I:C complexed with the lipofectamine 2000 transfection reagent to activate PKR (Appendix Fig. S4). Cell metabolites were isolated after six hours and then assessed by liquid chromatography coupled to mass spectrometry (LCMS). LCMS data acquisition and subsequent untargeted analysis (IDEOM (Creek et al, 2012)) identified 325 putative metabolites in the macrophage extracts, with a large number of significant differences between the *Eif2ak2*$^{-/-}$ and WT sample groups (Fig. 1A,B, data available at the Metabolomics Workbench, ID ST002412). A heatmap of the data reveals predominant suppression of metabolites in the WT compared to the *Eif2ak2*$^{-/-}$ cells, with depletion of

specific metabolites in amino acid, carbohydrate, lipid, and nucleotide pathways, while metabolites derived from the breakdown of amino acids were significantly elevated in the WT compared to the *Eif2ak2*$^{-/-}$ cells (Fig. 1A). The decreased levels of amino acids is accompanied by elevated intermediates in the urea cycle, suggesting amino acid catabolism. Phosphate energy metabolism is also altered with decreased creatine and phospho-creatine and a compensatory increase in phosphorylated guanidi-noacetate in the WT compared to the *Eif2ak2*$^{-/-}$ cells. These changes suggest a pseudo-starvation response in the WT cells. Pathway-level analysis revealed the four most substantially impacted were amino sugar and nucleotide sugar metabolism, arginine and proline metabolism, purine metabolism, and the PPP. Of these connected responses, the PPP had the highest impact (Fig. 1C). This is a multienzyme pathway that produces ribose sugars and reducing agents required for cell proliferation and survival.

The PPP consists of two decoupled phases. An aerobic phase oxidizes glucose 6-phosphate to produce carbon dioxide, NADPH, and ribulose 5-phosphate. Subsequent isomerization and epimer-ization in a nonoxidative phase convert the ribulose 5-phosphate with fructose 6-phosphate and glyceraldehyde 3-phosphate from the glycolytic pathway into different intermediates, the most important of which is ribose 5-phosphate, and returns unused carbon to the glycolytic/gluconeogenic pathway. Strikingly, all PPP intermediates detected are lower in the WT relative to the *Eif2ak2*$^{-/-}$ cells, except for xylose 5-phosphate (Fig. 1D). This exception is intriguing as the reversible activity of the enzymes at this point of the PPP should equilibrate the metabolites. The counterpart sugar at this position, ribose 5-phosphate, could not be distinguished from its precursor, ribulose 5-phosphate, under the employed LCMS conditions, although the levels of the combined sugars were also at odds with the pattern of the other metabolites in the pathway (Fig. 1D). The relative levels of biomolecules derived from ribose 5-phosphate were significantly lower in the WT compared to the *Eif2ak2*$^{-/-}$ cells (adenosine is shown in Fig. 1D), thereby supporting PKR-dependent suppression of the levels of ribose 5-phosphate. Accordingly, there appears to be reduced conversion of ribulose 5-phosphate to ribose 5-phosphate leading to the accumulation of this precursor and increased production of the surrogate pentose xylose 5-phosphate. This is in keeping with the decrease in the other PPP intermediates, fructose 6-phosphate, sedoheptulose 7-phosphate, and octulose 8-phosphate in the WT compared to the *Eif2ak2*$^{-/-}$ cells (Fig. 1D), which are produced by the transketolase or transaldolase using ribose 5-phosphate as a common constituent. The heightened production of fructose 6-phosphate in the absence of PKR has altered the ratio of glucose 6-phosphate to glucose due to carbon recycling through the upper gluconeogenic pathway in the *Eif2ak2*$^{-/-}$ cells (Fig. 1D) (Kuehne et al, 2015; Ralser et al, 2007). Notably, there is not a major difference in the intermediates lower in the glycolytic pathway that isn't shared with the PPP due to this carbon cycling. Juxtaposed with the reduction in the levels of most metabolites, the WT cells have increased levels of polyols (sorbitol, mannitol, and arabitol) relative to the *Eif2ak2*$^{-/-}$ cells (Fig. 1D). Heightened levels of these sugar alcohols have been reported in patients with impaired activity of the enzymes of the nonoxidative phase of the PPP (Boyle et al, 2016; Huck et al, 2004; Verhoeven et al, 2001). These data recognize PKR-dependent suppression of the nonoxidative phase of the PPP.

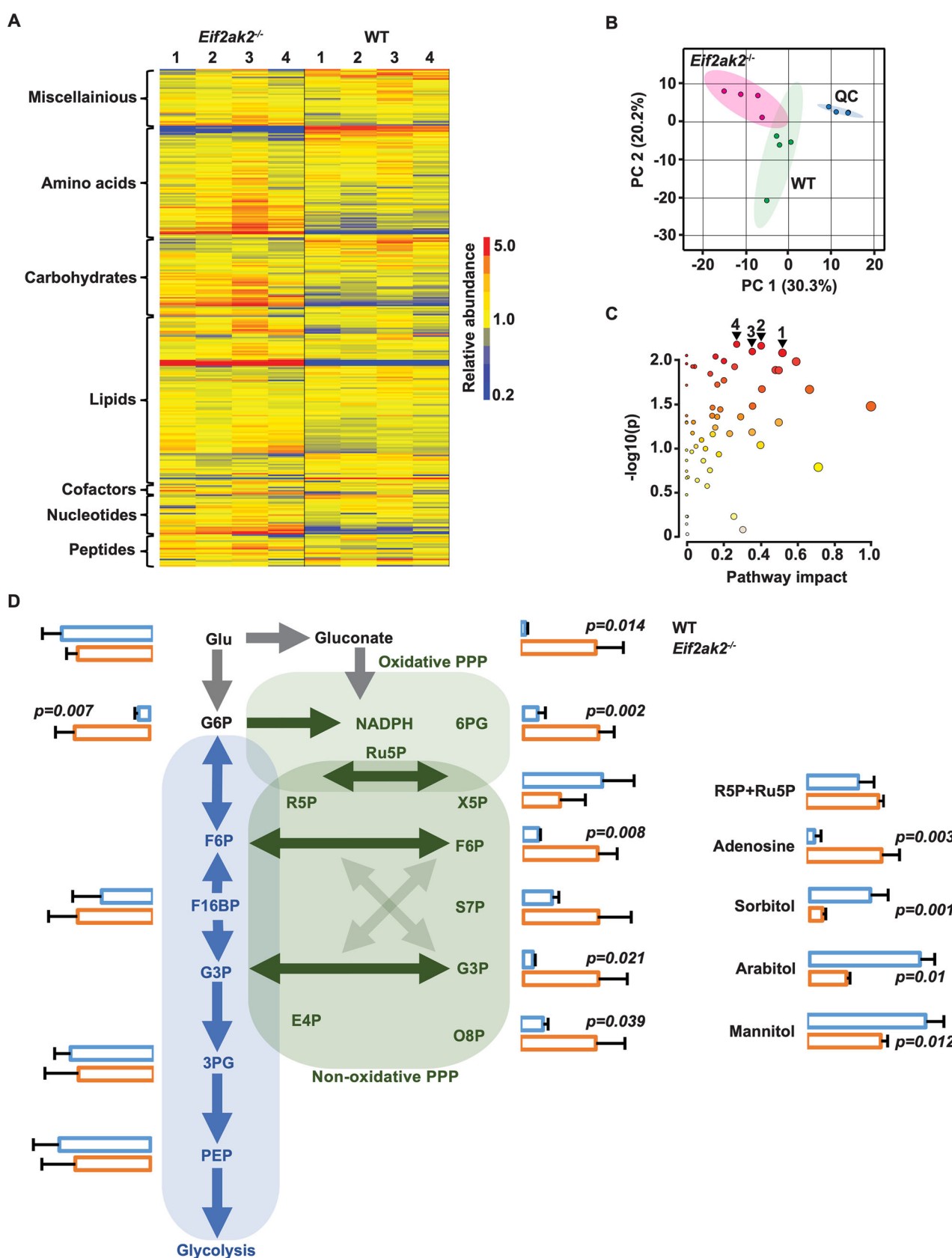

◀ **Figure 1. PKR alters glucose metabolism.**

An assessment of metabolites from WT and *Eif2ak2*$^{-/-}$ macrophages by LCMS. (A) A heatmap of metabolite levels clustered according to pathway map annotations in IDEOM. (B) Principal Components Analysis scores plot of metabolic profiles in WT (green) and *Eif2ak2*$^{-/-}$ (pink) cells relative to instrument quality controls (blue). (C) Pathway enrichment analysis by ANCOVA with relative-betweenness centrality topology analysis and a study-specific reference metabolome. The four most significant pathways are labeled; 1 = pentose phosphate pathway, 2 = arginine and proline metabolism, 3 = purine metabolism, and 4 = amino sugar and nucleotide sugar metabolism. (D) A schematic showing carbon metabolism and specific metabolite levels within central glycolysis (blue) and the PPP (green). The relative levels of metabolites in the WT and *Eif2ak2*$^{-/-}$ macrophages are depicted as horizontal bars (blue or orange, respectively). Abbreviations refer to; glucose (Glu), glucose 6-phosphate (G6P), 3-phosphoglyceric acid (3PG), phosphoenolpyruvate (PEP), fructose 6-phosphate (F6P), fructose-1,6-bisphosphate (F16BP), 6-phosphogluconate (6PG) glyceraldehyde 3-phosphate and/or dihydroxyacetone phosphate (G3P), ribulose 5-phosphate (Ru5P), ribose 5-phosphate (R5P), xylulose 5-phosphate (X5P), erythrose 4-phosphate (E4P), sedoheptulose 7-phosphate (S7P), and octulose 8-phosphate (O8P). Data were reported as the $\bar{x} \pm \sigma$. *P* values were calculated by the student's *t*-test (*n* = 4 independent experiments). Metabolic data were available at the Metabolomics Workbench (ID ST002412) and Source Data file 1. Source data are available online for this figure.

## PKR limits ribose 5-phosphate production

The preceding data identify that PKR alters the PPP, with the point of control appearing to be at the conversion of ribulose 5-phosphate to ribose 5-phosphate. Ribose 5-phosphate is a precursor for flavin adenine dinucleotides (FAD). Consistent with a restriction of ribose 5-phosphate production by PKR, there were lower FAD levels in the WT compared to the *Eif2ak2*$^{-/-}$ macrophages treated with poly I:C as measured by coupled enzyme assay and the coenzyme's intrinsic fluorescence captured by fluorescence-activated cell sorting (FACS) (Fig. 2A,B). This difference was repeated in a separate immortalized bone marrow-derived macrophage line expressing either a non-targeting control (coded shCONT) or *Eif2ak2*-targeting short hairpin RNA (shPKR, achieving 71% knockdown of the protein) (Fig. 2C; Appendix Fig. S4).

Ribose 5-phosphate is also a precursor for nucleotides, and we identify that PKR alters the levels of nucleic acids in the bone marrow-derived macrophages, as measured by fluorescent staining and FACS (Fig. 2D,E; Appendix Fig. S4).

The recycling of excess carbon produced in the nonoxidative phase of the PPP through the upper gluconeogenic pathway and back into the oxidative phase of the PPP in the *Eif2ak2*$^{-/-}$ cells obscures the point at which PKR controls the PPP. To assess the activity of the oxidative phase of the pathway, we measured the production of Nicotinamide adenine dinucleotide phosphate (NADPH) and its oxidized form (NADP+) in the splenic and bone marrow-derived macrophages by bioluminescent enzyme assay. This detects no effect of PKR on the nicotinamide nucleotides (Fig. 2F). The LCMS analysis also detected no significant difference in the nicotinamide nucleotide in the splenic macrophages, thereby supporting a point of control in the nonoxidative phase of the PPP.

To further probe the PPP, we rederived the isogenic macrophage lines to express; glucose 6-phosphate dehydrogenase (G6PD), 6-phosphogluconolactonase (PGLS), the transketolase (TKT) or the transaldolase (TALDO1). Expression of these PPP enzymes had no measurable effect on FAD fluorescence in the shCONT cells as measured by FACS (Fig. 2G). However, forced expression of the Ribose 5-Phosphate Isomerase A (RPIA) that produces ribose 5-phosphate increased fluorescence, while expressing an enzymatically impaired mutant RPIA with arginine replacement of a catalytic lysine residue (K173R (Zhang et al, 2003)) had no effect (Fig. 2G). Expression of RPIA in the PKR-ablated cells further increased the level of fluorescence (Fig. 2H). While forced

expression of TALDO1 reduced the heightened fluorescence of the PKR-ablated cells consistent with the enzymes' function to remove intermediates in the nonoxidative phase of the pathway to reduce the amount of ribose 5-phosphate for FAD (Fig. 2H). This alternating increase and decrease in FAD fluorescence through the expression of RPIA or TALDO1 supports a PKR-induced restriction of ribose 5-phosphate production.

The effects of PKR were further assessed by testing the capacity of the bone marrow-derived macrophages treated with poly I:C to use different carbon substrates. The ability of cells to metabolize glucose, glycine, D-ribose, D-xylose, L-arabinose, or D-sedoheptulose was tested over 48 h of culture by counting the cells. The latter four sugars can be taken up and phosphorylated by kinases to supply intermediates to the PPP (ribose 5-phosphate, xylulose 5-phosphate, and sedoheptulose 5-phosphate), while glucose and glycine are metabolized by both central glycolysis and the PPP (as glyceraldehyde 3-phosphate). Notably, the macrophages did not survive in a medium supplemented with the non-metabolizable glucose analog 2-deoxy-D-glucose. Also, the macrophages did not replicate appreciably without glucose supplementation but did persist to an extent on residual carbon sources in the fetal bovine serum (Fig. 2I). Together this identifies that these cells ostensibly require glucose. However, knocking down PKR expression (shPKR) increased the cell number in the absence of carbon supplementation, suggesting PKR activity exacerbates this dependency (Fig. 2J). Moreover, PKR expression differently affected the cell number upon the supplementation with different carbon sources. Supplementing the shCONT cells with ribose or sedoheptulose but not the other sugars reduced the limiting effect of PKR (Fig. 2J). This appears consistent with a PKR-dependent limitation of ribose 5-phosphate as ribose may be phosphorylated and sedoheptulose can be converted (with glyceraldehyde 3-phosphate) to produce ribose 5-phosphate. Moreover, combining ribose and xylose, to provide both transketolase substrates, increased the cell number over the single carbon source (Fig. 2J). These data are alternatively shown as the relative effect of supplementing each cell line (Appendix Fig. S4).

## PKR's control of the PPP is distinct from its recognized activity

Recombinant macrophage lines were produced that expressed components of the PKR response that were then assessed for their effect on FAD fluorescence. Forced expression of the established PKR substrate EIF2α or its guanosine diphosphate (GDP) exchange factor EIF2B, even as a mutant form (L335Q) reported to be insensitive to EIF2α phosphorylation (Bogorad et al, 2017), or the

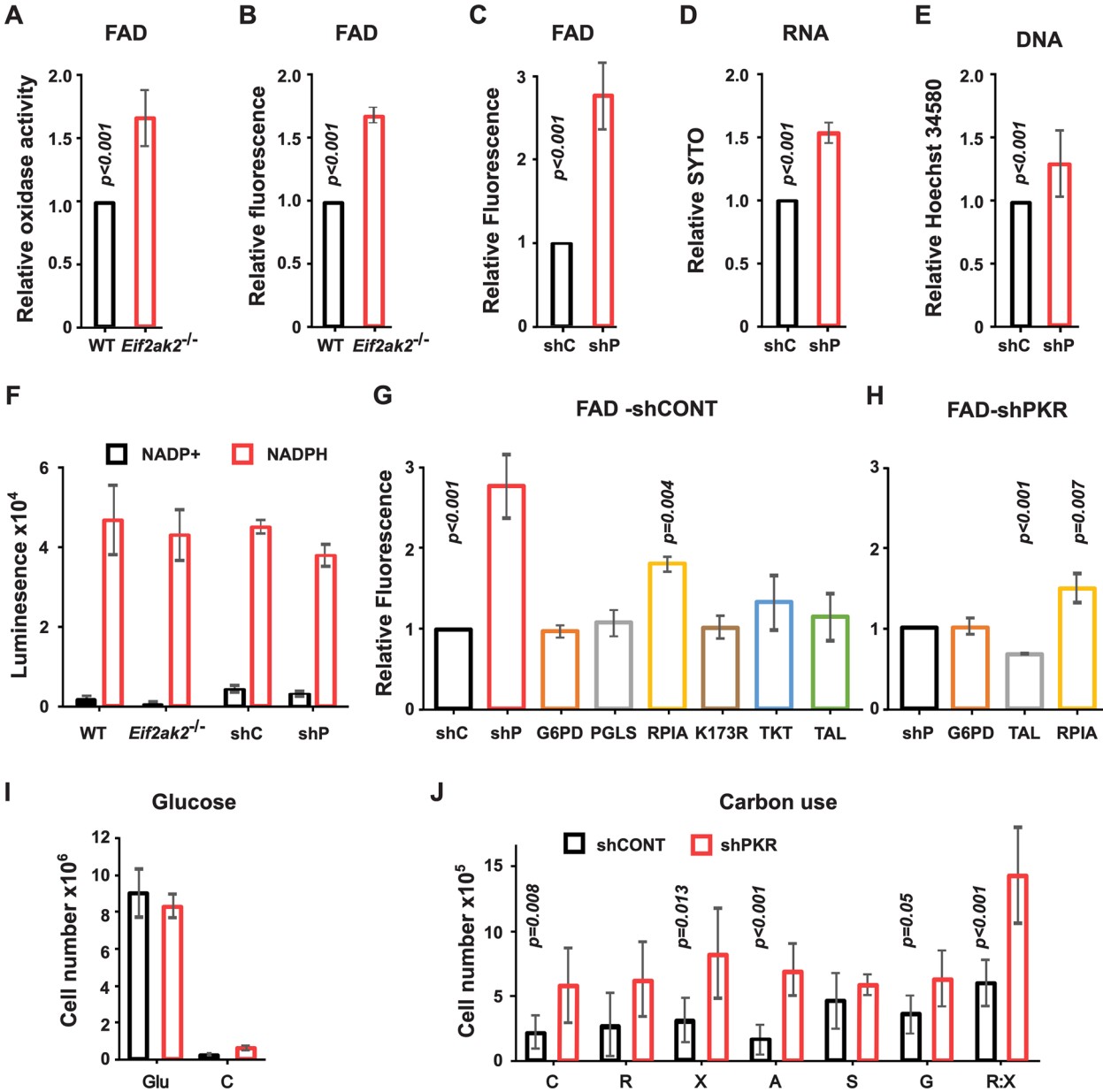

**Figure 2. PKR limits ribose 5-phosphate production.**

(A–F) Measures of the impact of PKR activity on the major products of the oxidative and nonoxidative phases of the PPP. (A–C) Plots of the effect of PKR on the levels of ribose 5-phosphate produced in the nonoxidative phase of the PPP by measures of FAD by (A) coupled enzyme assay ($n = 6$ independent assays) or (B) the molecules fluorescence in WT and *Eif2ak2*$^{-/-}$ splenic macrophages ($n = 4$) and (C) bone marrow-derived macrophages expressing the control (shC) or *Eif2ak2*$^{-/-}$ targeting (shP) shRNA as detected by the FACS ($n = 3$). (D, E) Plots of the effect of PKR on the levels of ribose 5-phosphate by measures of nucleic acids in the control (shC) or PKR-ablated (shP) bone marrow-derived macrophages through staining of (D) RNA with SYTO RNASelect ($n = 6$) or (E) DNA with Hoechst 34580 and detection by FACS ($n = 5$). (F) Plots of the effect of PKR on the levels of the reduced (NADPH) and oxidized (NADP+) forms of nicotinamide adenine dinucleotide phosphate produced in the oxidative phase of the PPP in the splenic WT and *Eif2ak2*$^{-/-}$ and the control (shC) or PKR-ablated (shP) bone marrow-derived macrophages by bioluminescent assay (NADP/NADPH-Glo assay) ($n = 3$). (G, H) Plots demonstrating the impact of expressing the indicated constructs on the PKR-dependent control of ribose 5-phosphate production by measures of FAD fluorescence in bone marrow-derived macrophages co-expressing a (G) non-targeting control (shCONT) or (H) *Eif2ak2*-targeting (shPKR) shRNA ($n = 3$). A catalytically impaired, point mutant RPIA construct is abbreviated as K173R. (I, J) Plots of cell counts after culture without the addition of saccharides (C) or supplemented with (I) glucose (Glu) ($n = 6$) or (J) D-sedoheptulose (S, $n = 3$), D-ribose (R, $n = 11$), D-xylose (X, $n = 6$), L-arabinose (A, $n = 6$) or glycerol (G, $n = 4$) alone and D-ribose combined with D-xylose (R:X, $n = 6$). The data show the $\bar{x} \pm \sigma$. P values shown were calculated by an unpaired student's t-test of independent experiments. Figures are derived from Source Data file 2. Source data are available online for this figure.

EIF2α phosphatase Protein phosphatase 1 regulatory subunit 15 A (Ppp1r15a) didn't alter the levels of FAD, as assessed by intrinsic cell fluorescence (Fig. 3A). Expression of a second phosphatase that is a PKR substrate, Protein Phosphatase 2 Regulatory Subunit Bα (Ppp2r5a or B56α), that had been reported to alter RPIA activity was also ineffective (Fig. 3A) (Ciou et al, 2015; Xu and Williams, 2000). Importantly, expression of a mutant *EIF2AK2* open reading frame (mutated at nucleotide positions *811AA > CG812*) to alter the catalytic lysine residue (K296R) did not restore the levels if FAD in cells expressing shRNA targeting the untranslated region of *Eif2ak2* (Fig. 3A, shPKR: shPKR-K296R, ratio 1.19, *p* = 0.87). This supports a requirement for PKR's kinase activity to control the PPP. Knockdown of the activating transcription factor 4 (ATF4) transcription factor, which is induced following EIF2α phosphorylation, significantly promoted cell fluorescence (Fig. 3A). This is consistent with the anabolic activities of ATF4, which is both EIF2α-dependent and -independent (Ben-Sahra et al, 2016; Duvel et al, 2010; Harding et al, 2000; Park et al, 2017). As there was not a marked additional contribution of ATF4 to PKR's control of FAD levels, this appears secondary to the kinase's control of the PPP (Fig. 3A, shCONT-shATF4: shPKR-shATF4, ratio 1.07, *p* = 0.56). Notwithstanding some involvement of ATF4, these data do not support a mechanism that encompasses PKR's established control of translation initiation.

Examination of the levels of the PPP gene transcripts in WT and *Eif2ak2*$^{-/-}$ splenic macrophages at rest and treated to activate PKR detected no major differences in their relative expression that account for the observed metabolic effects (Fig. 3B and Appendix Fig. S4). Measures of the levels of the gene products by immunoblot also detect no major effect on the expression of the enzymes by PKR activity (Fig. 3C and Appendix Fig. S4). Pharmacological targeting of the PKR response also didn't alter the expression of select PPP enzymes (Appendix Fig. S4). These data demonstrate that PKR's regulation of the PPP is not by altered gene expression via the kinase's established control of translation, by induction of the integrated stress response, or by altered cell signaling.

To identify the mechanism by which PKR suppresses the PPP, we undertook an unconventional approach that allowed us to unequivocally distinguish PKR's phosphorylation of EIF2α and its effects on cell signaling pathways in eukaryotic cells. Our approach was prompted by an observation made during recombinant protein expression in bacteria. Expression of the human WT PKR but not a kinase-dead PKR (K296R), repressed the growth of transformed *Escherichia coli* (Fig. 3D). This repressive effect of the kinase was also overcome by co-expressing the λ bacteriophage protein phosphatase (Fig. 3E), thereby identifying that PKR phosphorylates a bacterial protein that is required for growth. Unlike in eukaryotes, the methionine initiator binding is not required in *E. coli* for the ribosome to associate with mRNA and so translation initiation is differently controlled. Other PKR activities, such as the induction of the type I interferons, are also excluded in this context. Accordingly, this finding excludes PKR's orthodox activity.

To identify the putative bacterial substrate, we conducted a genetic complementation assay by mating *E. coli* strain BL21(DE3) that was transformed with an inducible (isopropyl beta-D-1-thiogalactopyranoside (IPTG)) human PKR with strains carrying mobile F-plasmids expressing different metabolic enzymes (Appendix Table S1 and Appendix Fig. S3). Ensuing culture under restrictive conditions for the exogenous genes identified that select enzymes from the PPP and the connected glycolytic and fatty acid synthesis pathways were able to restore growth. Horizontal gene transfer of phosphoglucose isomerase, 6-phosphofructokinase 1, fructose-1,6-bisphosphatase 1, pyridoxal phosphate/fructose-1,6-bisphosphate phosphatase, malate dehydrogenase, 3-hydroxy-acyl dehydratase, enoyl-acyl-carrier-protein reductase, β-hydroxy-acyl-acyl-carrier-protein dehydratase, 3-oxoacyl-acyl-carrier-protein synthase 1, acyl-carrier-protein S-malonyl transferase, acetyl-CoA carboxyltransferase subunit α, biotin carboxyl carrier-protein, biotin carboxylase, acetyl-CoA carboxyltransferase subunit β, Acyl-carrier-protein, 3-oxoacyl-acyl-carrier-protein synthase 3, ribose 5-phosphate isomerase B (RpiB), or Tkt all reduced kinase-dependent repression. This supports PKR activity inducing a metabolic check within the PPP and linked metabolic activity (Fig. 3F; Appendix Fig. S5).

To identify a possible point of control within the PPP, *E. coli* expressing WT or kinase-dead PKR were co-transformed with constructs of the *E. coli* PPP genes. Notably, as PKR is highly expressed in this experiment and is highly processive, rescue will be by metabolic restoration rather than by substrate overexpression allowing escape from phosphor-control. The growth of transformants under selective pressure for each construct was then monitored in liquid culture upon induction of PKR expression. Expression of enzymes from the oxidative phase of the PPP did not restore bacterial growth, thereby suggesting a point of control in the following nonoxidative phase of the pathway (Fig. 3G). Only the result for the first rate-limiting enzyme (Zwischenferment (Zwf)) of this phase is shown, although the effect of all the orthologous human enzymes is shown below. The expression of the enzymes from the nonoxidative phase restored growth, except Ribose 5-phosphate isomerase (RpiA), suggesting the isomerase as a PKR target (Fig. 3G; Appendix Fig. S6). Expression of RpiB, which can alternatively convert ribulose 5-phosphate from the oxidative phase of the PPP to ribose 5-phosphate, also rescued growth (Fig. 3G), thereby confirming that PKR suppresses ribose 5-phosphate production.

## PKR controls the ribose 5-phosphate isomerase

The human PPP enzymes were assessed in the same manner. RPIA with the enzymes from the oxidative phase was ineffective, while expression of TKT and the Transaldolase (TALDO1) from the nonoxidative phase restored bacterial growth (Fig. 4A; Appendix Fig. S7). Mutant forms of TKT (D155A) or TALDO1 (S171F) that were identified in patients with defects in PPP metabolites did not rescue growth (Fig. 4B) (Leduc et al, 2014; Wang et al, 1997). This confirms that the rescue of the PKR-induced auxotrophic effect is by metabolic compensation and verifies that PKR suppresses the production of ribose 5-phosphate by suppressing the activity of the human, with the analogous *E. coli*, isomerase. Notably, the ternary structures of the human and *E. coli* ribose isomerase enzymes are highly homologous, thereby supporting the potential for equivalent control by PKR (Appendix Figs. S8 and S11).

These genetic experiments were extended with metabolic experiments testing different carbon sources. The preceding experiments used glucose as a carbon source which is separately metabolized to provide multiple different sugars to the PPP (glucose 6-phosphate, fructose 6-phosphate, and glyceraldehyde 3-phosphate). The capability of *E. coli* to metabolize five-carbon

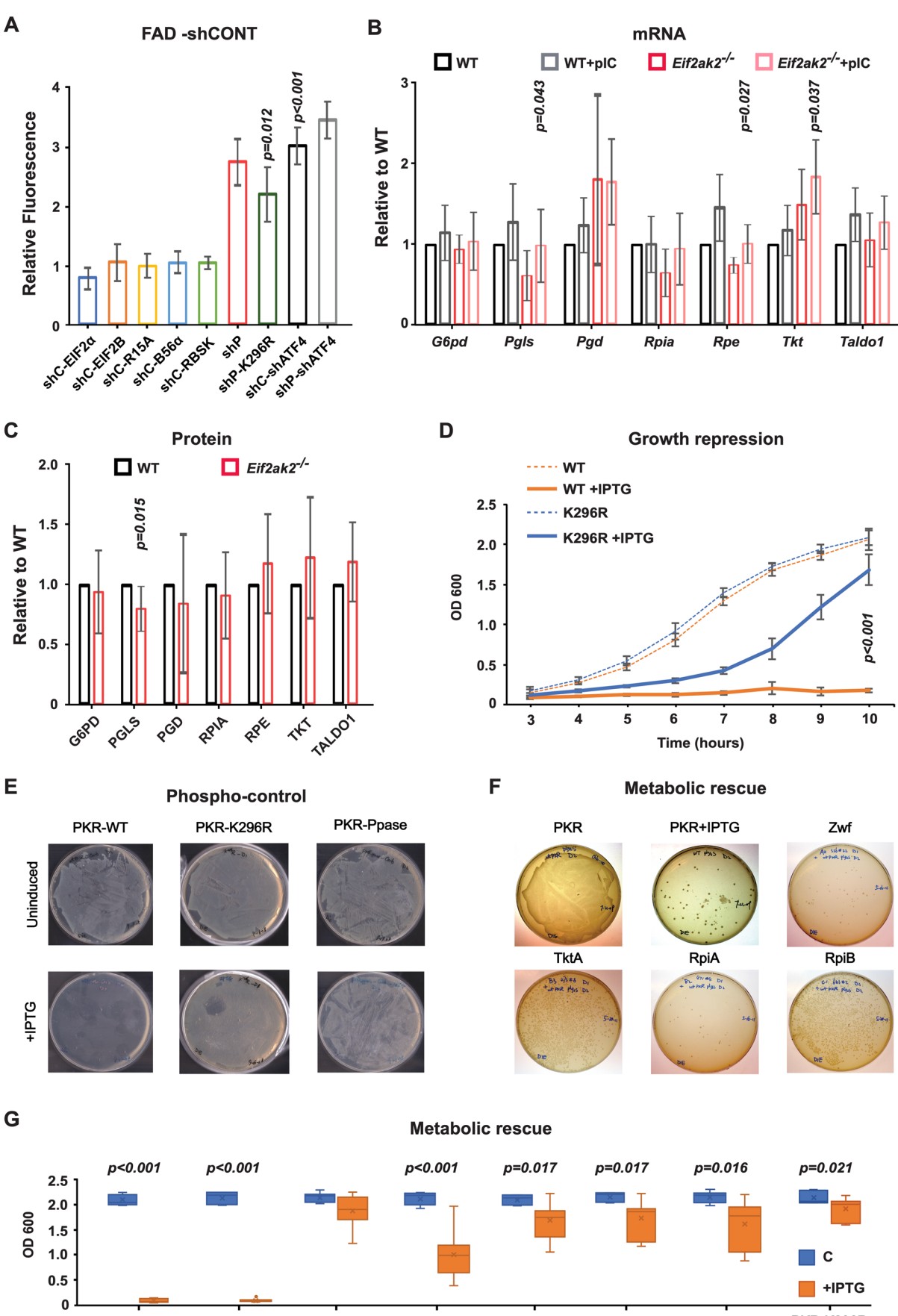

◄ **Figure 3. PKR controls the PPP independent of its orthodox activity.**

(A) A plot of the impact of expressing the indicated constructs on the PKR-dependent control of ribose 5-phosphate production by measures of FAD fluorescence in bone marrow-derived macrophages co-expressing shRNAs as a non-targeting control (shC) or targeting *Eif2ak2* (shP) or *Atf4* (shATF4) (*n* = 3). Abbreviations refer to Ppp1r15A (R15A), Ppp2r5a (B56α), Ribose kinase (RBSK), and the Eukaryotic initiation factor 2B (EIF2B). The probability values are calculated for the comparison to the control (shC). The shP and shP-shATF4 samples are also significantly different from the control (*p* < 0.001) but not their cognate shP-K296R and shC-shATF4 treatments (*p* = 0.899 and *p* = 0.071, respectively). (B) Measures of the PPP transcripts in WT and *Eif2ak2*$^{-/-}$ splenic macrophages by Q-PCR. Each gene is normalized to its level in the untreated WT sample (*n* = 3). The relative levels of the different transcripts are shown in Appendix Fig. S4. (C) Measures of PPP proteins in WT and *Eif2ak2*$^{-/-}$ splenic macrophages by quantitation of immunoblots detected with the indicated antibodies and normalized to the levels of the total protein detected by Ponceau S staining (*n* = 3–9). The data show the $\bar{x} \pm \sigma$. (D–G) Analysis of the effect of PKR's kinase activity on the growth of transformed *E. coli* upon inducing (+IPTG) expression of the WT or kinase-dead (K296R) PKR transgenes in (D) liquid culture (*n* = 3) or (E) on solid medium, also by co-expression of the bacteriophage λ protein phosphatase (Ppase) with the WT PKR (representative images). (F) Representative images of bacterial colonies on culture plates arising from transformed bacteria expressing PKR and the indicated *E. coli* genes transferred by genetic complementation from donor strains carrying the factors as mobile genetic elements. All bacterial genes horizontally transferred to *E. coli* transformed with PKR are listed in Appendix Table S1, and representative images of those that rescued the PKR-dependent repression of growth are shown in Appendix Fig. S5. (G) Plots of bacterial growth measured by the optical density (at 600 nm) of *E. coli* co-transfected with kinase-dead (K296R) or WT PKR, uninduced or induced with IPTG and the indicated enzymes. The growth of the matched K296R controls for all genes is shown in Appendix Fig. S6. The data is presented as box plots in which the box marks the first and third quartiles, with an additional line marking the median, and the minimums and maximums outside the first and third quartiles depicted with whiskers. *P* values shown were calculated by an unpaired student's *t*-test of independent experiments (*n* = 7). Figures are derived from Source Data file 3. Source data are available online for this figure.

sugars via the PPP was used to identify the specific metabolic deficiency induced by PKR. Transformed *E. coli* were cultured with arabinose, xylulose, or ribose. Each of these pentoses is taken up and phosphorylated by the respective prokaryotic kinases to supply ribulose 5-phosphate, xylulose 5-phosphate, or ribose 5-phosphate at the point in the PPP that appeared to be controlled by PKR (Fig. 4C). Use of any of these sugars as a sole carbon source requires conversion by both the isomerase (RpiA) and epimerase (Ribulose 5-phosphate 3-epimerase (Rpe)) to supply the counterpart substrate to the downstream transketolase. Without this conversion, the single carbon source cannot be metabolized. While *E. coli* expressing the kinase-dead PKR grew efficiently on each of these single carbon sources, bacteria expressing the active kinase did not (Fig. 4D). This confirms that the phosphor-control by PKR inhibits the conversion of ribulose 5-phosphate. Providing complementary substrates by combining arabinose or xylose with ribose, thereby negating the requirement for the isomerase or epimerase, overcame the kinase-induced auxotrophic effect (Fig. 4D). This specific metabolic block with the preceding genetic rescue confirms that PKR suppresses the production of ribose 5-phosphate by inhibiting the activity of the isomerase within the nonoxidative phase of the PPP.

## Phosphor-control of the ribose 5-phosphate isomerase

We sought to identify if PKR directly phosphorylates the ribose isomerase. To assess the phosphorylation of human RPIA in the bacterial assays, we immune-enriched the FLAG-tagged RPIA with an anti-FLAG antibody from the lysates of *E. coli* co-expressing either the WT or kinase-dead PKR. Immunoprecipitated proteins were resolved by gel electrophoresis and then visualized with phosphoprotein and total protein stains. The later total protein stain shows *E. coli* proteins are copurified in the immune-enriched fraction and detects minor variance in the amounts recovered (Fig. 5A). The phosphoprotein stain identifies that several of the prokaryotic proteins are phosphorylated by endogenous kinases (as this is independent of PKR's kinase activity, Fig. 5A). These *E. coli* phosphoproteins serve as an internal control to verify that there is sufficient protein loaded in all samples to be able to detect phosphorylated RPIA. Two phosphoproteins coinciding with the expected molecular weights of RPIA isoforms are detected in the

lysates from cells expressing the WT PKR (Fig. 5A with C,D). This supports PKR-dependent phosphorylation of RPIA in the bacterial experiment. Although phosphorylated RPIA bands are absent from the lysates expressing the kinase-dead PKR, a faint phosphoprotein is detected that is coincident with the lower molecular weight RPIA peptide. This phosphoprotein was also detected in lysates from separate experiments that didn't express RPIA, independent of PKR's kinase activity (Appendix Fig. S3), and so could be a product of a prokaryotic kinase or even the *E. coli* isomerase, which is a phosphoprotein of this molecular weight, that might be co-enriched with the FLAG-RPIA (hetero-oligomerization is detected in Appendix Fig. S11 and the ternary structures are compared in Appendix Fig. S8) (Hansen et al, 2013; Lin et al, 2015).

To detect phosphorylation of the endogenous isomerase, the spleen-derived macrophages were treated with poly I:C to activate PKR, then lysed, and the proteins resolved by isoelectric focusing, followed by SDS-PAGE separation and transfer to membranes for immunoblotting with specific antibodies. This procedure can resolve different phosphorylation states of proteins by their altered charge. While suppression of phosphatase activity and enrichment of phosphoproteins is usually necessary to capture this transient modification, differently charged isoforms of RPIA were evident as low abundant fractions in the WT and *Eif2ak2*$^{-/-}$ cells (Fig. 5B). RPIA was detected as two different molecular weight proteins, with the larger peptide the major isoform, appearing as multiple subfractions in the pH gradient used for isoelectric focusing (Fig. 5B with C,D). Notably, a single protein fraction in the lysates from the *Eif2ak2*$^{-/-}$ cells refocus as two more negative fractions in the lysates from the WT cells (Fig. 5B). This is consistent with the addition of negatively charged phosphoryl groups to this protein isoform in the WT cells, thereby supporting its phosphorylation by PKR. No phosphorylation differences were detected in the other PPP enzymes tested (Fig. 5B).

To more conclusively test if the enzymes of the PPP are substrates for PKR, we conducted an in vitro kinase assay. Recombinant PPP proteins or, as a positive control, EIF2α were tagged at their carboxyl terminus with the FLAG epitope and expressed, then immune-enriched with an anti-FLAG antibody from HEK293 cells in which PKR expression was suppressed by RNA interference to reduce background phosphorylation. Although RPE was not able to be expressed satisfactorily, the

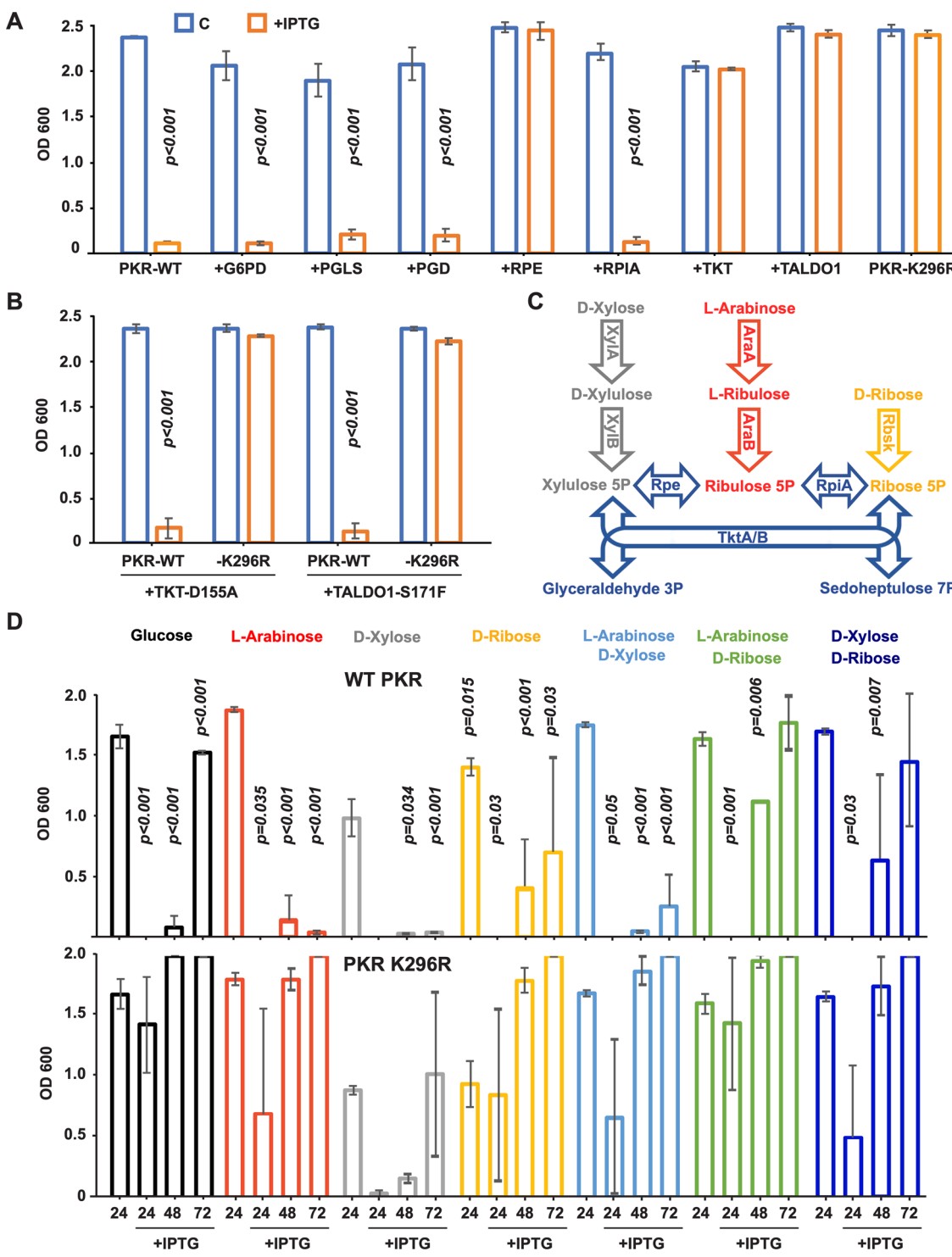

**Figure 4.   PKR controls the activity of the ribose 5-phosphate isomerase.**

(A) Plots identifying PKR suppresses RPIA activity by measures of the optical density (600 nm) of *E. coli* transfected with WT or kinase-dead (K296R) PKR and co-transfected with the indicated human PPP genes ($n = 3$). (B) Plots of the effect of the indicated catalytically impaired mutants of the human transketolase (TKT-D155A) or Transaldolase (TALDO1-S171F) on the growth of *E. coli* co-transfected with WT or kinase-dead (K296R) PKR confirming metabolic activity is required for the genetic rescue ($n = 3$). The growth of the matched K296R controls for all PPP genes is shown in Appendix Fig. S7. (C) A schematic of the metabolism of the indicated saccharides in *E. coli*. (D) Plots identifying that PKR's kinase activity controls the conversion of ribulose 5-phosphate to ribose 5-phosphate by measures of the optical density (600 nm) of *E. coli* transfected with WT (at top) or the kinase-dead (K296R) PKR (below) grown with the indicated carbon sources ($n = 3$–7). All plots show the $\bar{x} \pm \sigma$ of independent experiments. *P* values were calculated by unpaired student's *t*-test of independent experiments. Figures are derived from Source Data file 4. Source data are available online for this figure.

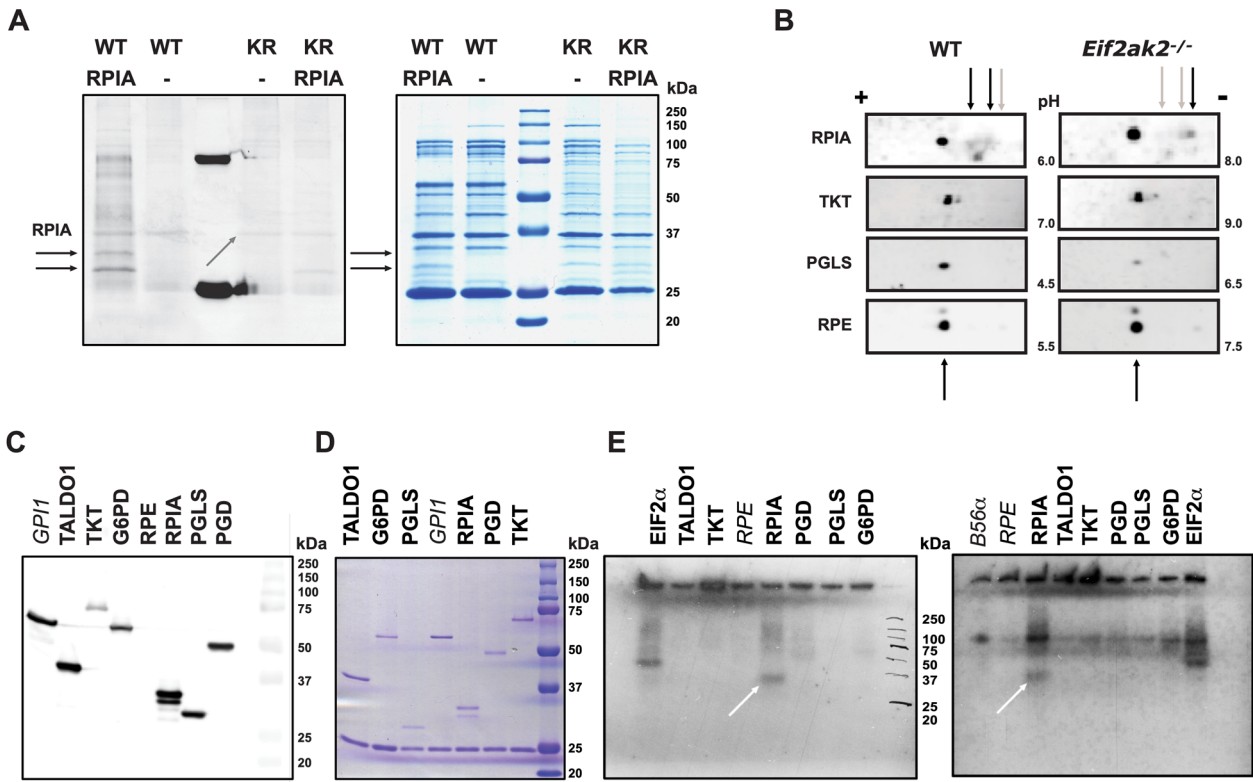

**Figure 5. PKR phosphorylates RPIA.**

(A) Identification of the phosphorylation of RPIA by PKR in bacteria by staining of phosphoproteins (Pro-Q™ Diamond on the left) and total proteins (Coomassie on the right) immunoprecipitated with an anti-FLAG antibody from *E. coli* expressing the kinase-dead (KR) or WT PKR alone or with FLAG-tagged RPIA, then separated by SDS-PAGE electrophoresis. A ubiquitous 37 kDa bacterial phosphoprotein is indicated with arrows within the stained gel as evidence that there are sufficient protein levels in all samples to be able to detect phosphorylated RPIA (indicated by arrows outside the gel). (B) Detection of the phosphorylation of endogenous RPIA by PKR through measures of differently charged species of the indicated proteins detected by immunoblotting of isoelectric focused lysates from WT and *Eif2ak2*$^{-/-}$ splenic macrophages. All the proteins exist as a major isoform that is equivalent between the two cell lines (indicated with an arrow below the immunoblots) except RPIA, which has additional minor isoforms that differ between the WT and *Eif2ak2*$^{-/-}$ cells (indicated by arrows above the immune blot). The local potential of the hydrogen gradient is indicated on the leftmost immunoblots. (C) An immunoblot with an anti-FLAG antibody detecting the expression of FLAG-tagged PPP proteins in HEK293 cells. (D) A Coomassie-stained SDS-PAGE gel visualizing the recovery of the indicated FLAG-tagged products from HEK293 cells by immunoprecipitation with an anti-FLAG antibody. (E) Autoradiographs of replicated experiments detecting phosphorylation of the indicated FLAG-tagged PPP proteins as γP$^{32}$-ATP products labeled by PKR's kinase activity then recovered by immunoprecipitation with an anti-FLAG antibody and resolved by SDS-PAGE. Phosphorylated RPIA is indicated by a white arrow in each autoradiograph. The established PKR substrates, EIF2α and B56α, were included as controls, although the latter is only indirectly evident by its stimulatory effect on PKR's auto-phosphorylation in the right autoradiograph as this HIS-tagged substrate was not immune-enriched. This figure used Source Data file 5. Source data are available online for this figure.

remaining PPP enzymes were isolated in this way (Fig. 5C,D). Recombinant PKR and a second control substrate PPP2R5A (also termed B56α (Xu and Williams, 2000)) were expressed in *E. coli* as a hexa-histidine-tagged protein and purified by Ni-agarose affinity chromatography. Phosphorylation of the PPP proteins by PKR's kinase activity was detected by radiolabeling with γP$^{32}$-ATP, electrophoretic separation of the reactants, followed by their visualization by autoradiography. We used an established assay protocol developed to test PKR activity modified to avoid the concealment of substrates by radiolabeled auto-phosphorylated PKR (Hovanessian et al, 1983). This involved immunoprecipitation of the reactants with an anti-FLAG antibody to enrich the target proteins relative to radiolabeled PKR. A vestige of auto-phosphorylated PKR is evident as a full-length 68 kDa and a cleaved 45 kD peptide in one of the two autoradiographs shown (Fig. 5E). These phosphoproteins coincide with TKT (68 kDa) and

could also potentially obscure PGD (49 kDa) (Fig. 5C–E). The labeled products from kinase assays identify RPIA as a previously unknown PKR substrate (Fig. 5E; Appendix Fig. S9). The direct labeling of the substrates is also indirectly supported by an observed stimulation of PKR's auto-phosphorylation (Fig. 5E).

These data demonstrate that the suppression of RPIA activity is accompanied by its phosphorylation by PKR.

## Identification of phosphoresidues on RPIA

To identify specific residues on RPIA that are phosphorylated by PKR, we evaluated a complex of the two proteins enriched from cells by mass spectrometry. Towards this, we constructed an orthogonal tag using a split fluorophore. Separate halves of the Venus fluorophore were tagged to each protein partner, and the constructs were co-expressed in HEK293 cells in which endogenous

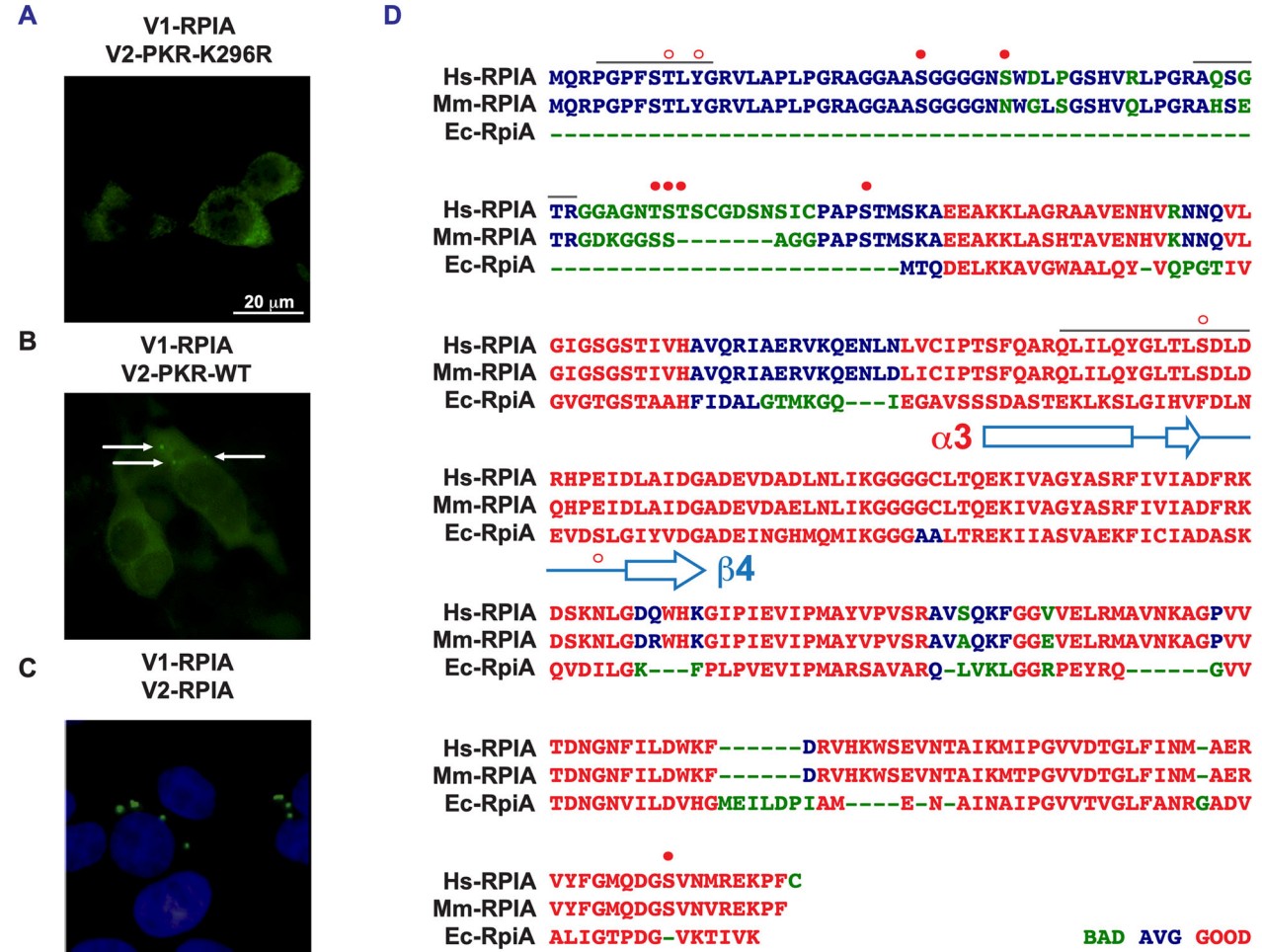

**Figure 6. Identification of PKR substrate residues on RPIA.**

(A–C) Micrographs demonstrating the spatial pattern of the colocalization of PKR and RPIA as a fluorescence signal generated through reassembly of a split Venus fluorophore (the amino and carboxyl fragments are labeled V1 or V2) separately attached to the protein partners and co-expressed in HEK293 cells. Venus-tagged constructs of RPIA are shown co-expressed with (**A**) kinase-dead (K296R) PKR, (**B**) WT PKR, or (**C**) itself. The arrows identify faint foci in the cytoplasm of cells expressing the split Venus RPIA with WT PKR. Fluorescence induced by the oligomerization of RPIA is discerned to be cytosolic by visualizing the cell nuclei with Hoechst 33342. (**D**) An alignment of the amino acid sequences of the ribose isomerase from humans (Hs), mice (Mm) and *E. coli* (Ec) to demonstrate the conservation and specific features. Phosphorylated residues correlated with PKR activity are indicated with solid red spots. Other putative phosphoresidues are indicated with a red circle above the human and below the *E. coli* sequences. An α3-β4 loop structural element predicted to be important in the interaction between RPIA and PKR is shown diagrammatically below the *E. coli* sequence. Potential phosphopeptides that weren't captured in the mass spectrometric analysis are indicated with a line above the human RPIA sequence. The amino acid sequence of the split Venus-tagged proteins is shown in Appendix Fig. S10. The conformations of the human and *E. coli* isomerases are compared in Appendix Fig. S8. The figure used Source Data file 6. Source data are available online for this figure.

PKR expression was suppressed by RNA interference (Liu et al, 2014). Colocalization of the tagged proteins reconstitutes Venus, which was then immune-enriched with an antibody that is specific for the full-length fluorophore. RPIA was tagged with the amino-terminal half of a split Venus (coded V1), then co-expressed with either the WT or kinase-dead PKR tagged with the complementary carboxyl-terminal split Venus peptides (coded V2) (Appendix Fig. S10). The fluorescent signal produced by the colocalization of these constructs in cells appears to detect a difference in the interaction between RPIA and the WT compared with the kinase-dead PKR (Fig. 6A,B). This may be a consequence of the relative expression levels, due to suppression of translation, or differences in the conformations of the active and inactive kinases (Dar et al, 2005;

Dey et al, 2005; Mayo et al, 2019). Notably, visualization of RPIA multimerization with split Venus-tagged constructs shows foci that appear more reflective of the fluorescent puncta that formed with the WT PKR rather than the diffuse pattern produced with the kinase-dead PKR (Fig. 6C) (Essenberg and Cooper, 1975; Rutner, 1970).

Split Venus-tagged human RPIA that was immunoprecipitated with either the WT or the kinase-dead PKR was digested with proteases and phosphoresidues were identified by mass spectrometry (data available via the PRIDE repository ID PXD036779). The amount of RPIA in the WT PKR complex was less abundant than in the kinase-dead PKR complex as assessed by spectral counting (800 vs. 1250 MS² spectra, respectively), consistent with

**Table 1. Detection of phosphorylated residues on RPIA.**

| Phosphorylated peptide detected | Residue modified | # MS$^2$ spectra WT | # MS$^2$ spectra K296R | Byonic score (WT) | Mass error (ppm) | P value |
|---|---|---|---|---|---|---|
| RPIA | | | | | | |
| R.AGGAAS$^p$GGGGNS$^p$WDLPGSHVR.L | S27 | 21 | 3 | 787.2 | 1.1 | $1.15 \times 10^{-13}$ |
| | S33 | 8 | 0 | 989.8 | 0.9 | $1.74 \times 10^{-18}$ |
| R.GGAGNT$^p$S$^p$T$^p$SCGDSNSICPAPS$^p$TMSK.A | T58 | 6 | 0 | 227.0 | 0.8 | $3.3 \times 10^{-5}$ |
| | S59 | 1$^a$ | 0 | 126.3 | 0.1 | $1.48 \times 10^{-3}$ |
| | T60 | 1$^{a, b}$ | 0 | 255.5 | 0.5 | $3.72 \times 10^{-5}$ |
| | S73 | 2$^a$ | 0 | 156.2 | 0.1 | $1.15 \times 10^{-3}$ |
| R.VYFGMQDGS$^p$VNMR.E | S302 | 3$^b$ | 0 | 42.7 | 0.7 | 0.026 |
| PKR | | | | | | |
| R.T$^p$R.S | T446 | 14 | 0 | 1107.6 | 0.8 | $8.71 \times 10^{-22}$ |

Phosphorylated amino acids were detected by tandem mass spectrometry of split Venus-tagged RPIA enriched with either the WT or kinase-dead (K296R) PKR (the amino acid sequences are shown in Fig. 6D; Appendix Fig. S10). Data of the highest-scoring peptide with the specific phosphorylated residue as the sole modification is shown unless indicated. Detection of the regulatory auto-phosphorylated threonine residue number 446 within the activation segment of the kinase is shown for reference.

[a,b] Peptides also contained cystine ([a]) or methionine ([b]) modifications.

lower protein levels (Appendix Fig. S10). The majority of RPIA was captured in the analysis (93% of the protein), although three peptides with potential phosphoresidues (serine, threonine, and tyrosine) had low or no coverage. Two of these peptides (amino acid sequence PGPFSTLYGR and AQSGTR) are within an amino-terminal extension that is missing from the prokaryote isomerase, while the third (QLILQYGLTLSDLDR) is within the core of the enzyme (Fig. 6D). A single phosphoserine at position 27 was identified on RPIA that was copurified with the kinase-dead PKR. This same residue with six others, S33, T58, S59, T60, S73, and S302 was identified as being phosphorylated on RPIA copurified with the WT PKR (Table 1; Fig. 6D). Five of these phosphoresidues had not previously been reported, while two were previously identified (S27 and S33) with others that are not detected here (T9, Y11, S39, S104, and S106, PhosphoSitePlus (Hornbeck et al, 2015)). Two of these latter phosphoresidues are located on a peptide that was not captured in our analysis (PGPFST$^p$LY$^p$GR) and so cannot be excluded as possible PKR phosphoresidues. Together eight residues appear as potential PKR phosphorylation sites.

## The function of phosphoresidues on RPIA

Computational modeling was used to support the phosphorylation analysis. For this, we used the established structure of the human PKR (PDB: 2A19) and a model of human RPIA based on the established structure of the E. coli isomerase produced by SWISS-MODEL (Studer et al, 2021). Protein docking and molecular superposition were used to predict how PKR and RPIA interact (Hex 8.0.0 (Ritchie and Kemp, 2000)). This predicts an orientation of the two proteins that conforms to known modes of interaction of each protein (Fig. 7A) (Capriles et al, 2015; Dar et al, 2005; Graille et al, 2005; Ishikawa et al, 2002; Zhang et al, 2003). Notably, RPIA is predicted to interact with PKR's substrate-binding residues in the αG-helix. In this orientation, a loop between the α3-helix and β4-strand elements of RPIA (referencing the E. coli RpiA structure (Zhang et al, 2003)) projects into the catalytic pocket of PKR. These residues were absent from the mass spectrometric data but encoded a potential phosphoserine residue at position 147 (Figs. 7A and

6D). In keeping with the effect of PKR in E. coli, the bacterial isomerase encodes an equivalent potential phosphoserine residue at position 75 in this α3-β4 loop (Fig. 6D). As the serine at position 147 on RPIA or 75 on E. coli RpiA present as feasible PKR substrates (Cho et al, 2020; Uppala et al, 2018), this position was assessed together with the preceding phosphoresidues identified by mass spectrometry for their effect on isomerase function.

Target residues were assessed by mutation and then testing in E. coli, as performed previously. Notably, the amino-terminal 74 amino acids of human RPIA are highly variable between isomerases from different species. These residues are not predicted to contribute to the core structure of the enzyme (as envisaged in the AlphaFold identifier AF-P49247-F1). Importantly, this region is absent from the E. coli enzyme and so cannot account for the PKR-dependent control in this context. Accordingly, the significance of phosphorylation of the amino-terminal extension of RPIA is uncertain. Rather than separately mutating each of the amino-terminal phosphoresidues, we deleted the first 74 amino acids of the human isomerase (coded δ74). The remaining phosphoresidues were replaced with alanine. RPIA constructs were expressed and immune-enriched from HEK293 cells, then their activity was assessed by in vitro spectrophotometric measures of the conversion of ribose 5-phosphate to ribulose 5-phosphate. Comparisons to the WT and a mutant enzyme with arginine replacement of a catalytic lysine residue at position 173 (K173R) show that there was no change to the isomerase activity of the constructs (Fig. 7B). Expression of the amino-terminal truncated RPIA (δ74) didn't restore the growth of E. coli expressing WT PKR (Fig. 7C). Neither did alanine replacement of the carboxyl-terminal phosphoserine (S302A) on RPIA or a corresponding threonine residue number 210 on RpiA (Figs. 7C and 6D). However, alanine replacement of the serine residue within the α3-β4 loop of RPIA (S147A) did afford a modest recovery (Fig. 7C). This recovery was replicated with an equivalent mutation (S75A) of the bacterial isomerase (Fig. 7D). These data identify the involvement of the serine residue within the α3-β4 loop of the isomerase in the PKR-dependent suppression. However, the scale of the recovery demonstrates prevailing suppression. This may stem from repressive phosphorylation at

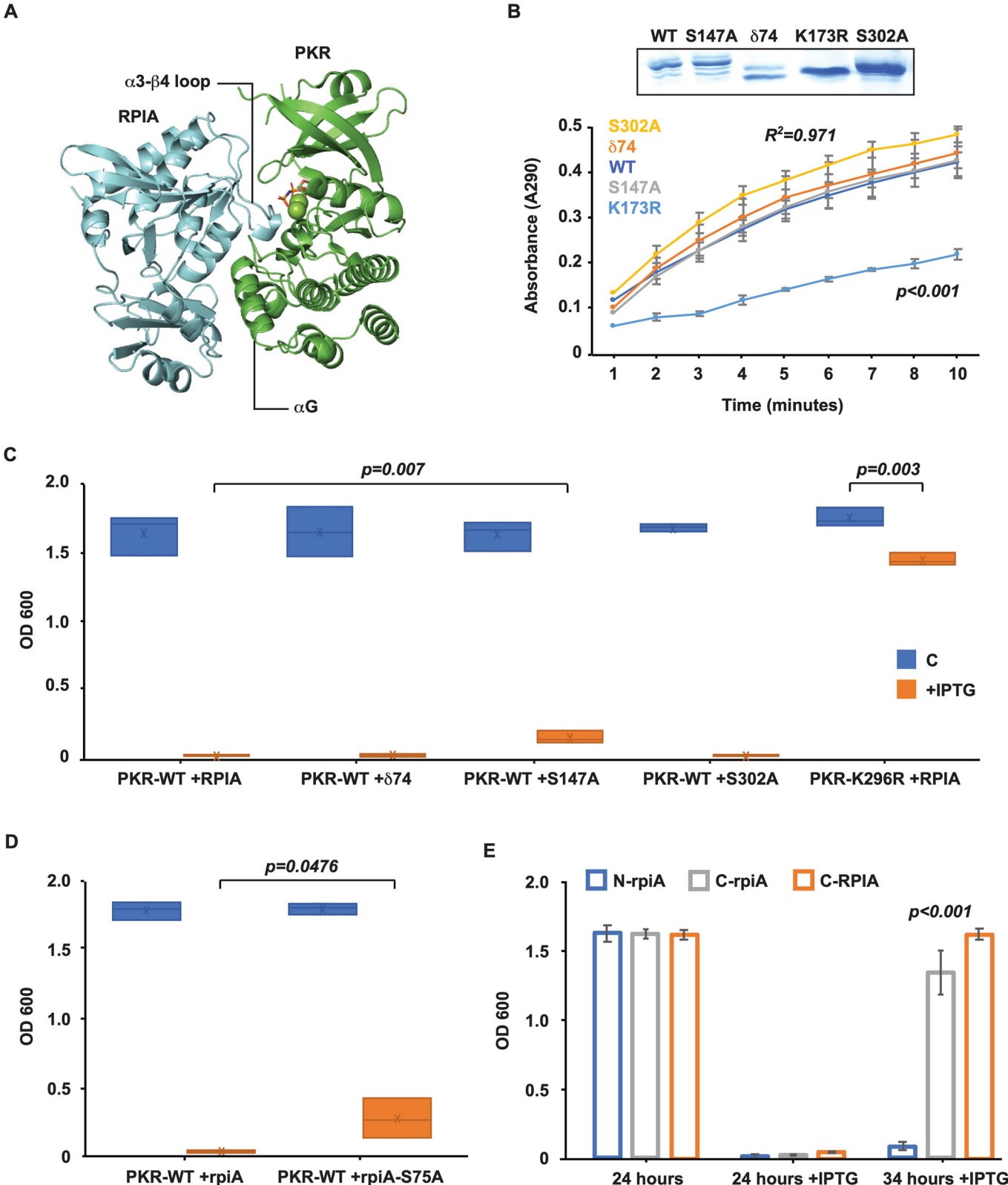

**Figure 7.   Functional assessment of RPIA phosphoresidues.**

(A) A ribbon model of the predicted interaction between RPIA (cerulean) and the kinase domain of PKR (green), with the bound ATP in stick form and magnesium ions as spheres. The substrate-interacting element of PKR (the αG-helix) engages RPIA in an orientation in which a loop between the α3-helix and β4-sheet on the isomerase projects into the catalytic pocket of the kinase. (B) Spectrometric measures of the activity of the indicated RPIA constructs. Preparations of the constructs are visualized by Coomassie staining of the SDS-PAGE separated proteins (inset). Isomerase activity is compared as a t-statistic of the slopes of the trend lines of the data ($n = 3$). The estimated goodness-of-fit ($R^2$) of the WT RPIA data is shown. (C–E) Plots demonstrating the impact of RPIA phosphoresidues by measures of the growth of *E. coli* expressing WT or kinase-dead (K296R) PKR alone or in conjunction with the indicated constructs of (C) human RPIA and (D) *E. coli* RpiA untreated or induced (+IPTG) at 37 °C with shaking for 24 h ($n = 3$). (E) Plots demonstrating that the carboxyl (C) but not the amino (N)-terminus of the isomerase relieves PKR-dependent suppression of bacterial growth after 34 h of culture ($n = 3$). The data show the $\bar{x} \pm \sigma$ of independent experiments. $P$ values were calculated by unpaired student's $t$-test of independent experiments. Figures are derived from Source Data file 7. Source data are available online for this figure.

other sites on the isomerase or, as the active enzyme is an oligomer (Fig. 6C; Appendix Fig. S11), interference from the endogenous phosphorylated isomerase or, as PKR is highly expressed in this experiment, an inhibitory interaction with the kinase without phosphorylation.

During the production of these constructs, we identified unintended frame-shift mutations that resulted in the expression of a truncated isomerase. Despite being enzymatically incompetent, these peptides increased bacterial growth after extended culture times. To confirm this was due to the partial isomerase sequence and not a divergent effect of a frame-shifted peptide, we subcloned sequences of the isomerase that expressed only the amino-terminal 74 or just the carboxyl-terminal 82 amino acids (beginning at M138) of the *E. coli* RpiA or the corresponding 92 carboxyl-terminal amino acids of human RPIA (beginning at M220) (Fig. 6D). Co-expression of the carboxyl-terminal fragments of either isomerase, but not the amino-terminal residues of RpiA, reduced the PKR-dependent repression of growth after prolonged culture (Fig. 7E). As our modeling predicts that RPIA interacts with PKR via its carboxyl-terminal residues (Fig. 7A), this peptide may relieve repression by competing with the endogenous isomerase for PKR.

## Pharmaceutical targeting of the PPP as an antiviral therapy

The function of PKR in the antiviral response suggests kinase control of the activity of the ribose isomerase would limit the production of nucleotides during infection to suppress viral replication. To test this concept, we measured the consequence of pharmacologically targeting this response during virus infection.

Previous studies have identified inhibitors of isomerases from plants and prokaryotes. Because these molecules were pseudo substrates and because the enzyme is highly conserved (Fig. 6D; Appendix Fig. S8), these inhibitors should also control the human enzyme. This prediction was tested using the 4-phospho-D-erythronate inhibitor of the prokaryotic RpiA (Roos et al, 2005; Zhang et al, 2003). We synthesized a derivative of 4-phospho-D-erythronohydroxamic acid (coded JNB-hu20-057-2A) and validated its activity against the human isomerase in a continuous spectrophotometric assay. This shows the inhibitor suppresses the activity of human RPIA as reported previously for isomerases from other species (Fig. 8A–C, *Spinacia oleracea* RpiA, Ki = 29 μM, Km/Ki = 260 (Burgos and Salmon, 2004)).

The charged nature of this inhibitor, as a phosphorylated sugar analog, precludes its uptake by cells, and so we resynthesized a cell-permeable form, again, based on a previously produced molecule (Ruda et al, 2007). This prodrug, coded as JNB-hu20-051A, was

assessed in murine embryonic fibroblasts (MEFs). The murine isomerase is almost identical to the human isomerase (Fig. 6D). A dose of 200 μM of JNB-hu20-051A was used, based on a binding affinity in the micromolar range and the previously demonstrated tolerance to a similar molecule (Ruda et al, 2007). Consistent with suppressed ribose 5-phosphate production with the ensuing rate of nucleic acid synthesis, the prodrug reduced the replication of the immortalized cell line after 48 h (Fig. 8D). No toxicity was evident, as assessed by vital dyes (Fig. 8E).

The antiviral activity of JNB-hu20-051A was tested against infection with the Herpes simplex virus (HSV-1). HSV-1 has a well-characterized mechanism that represses PKR activity to enable its replication (He et al, 1997; Leib et al, 2000; Poppers et al, 2000). We used a recombinant virus that expressed a GFP reporter to be able to visualize the course of infection (Elliott and O'Hare, 1999). As comparisons, MEFs were treated with the solvent ($H_2O$) or an approximate clinical dose (100 IU/mL) of the antiviral type I interferon β (IFNβ) cytokine applied an hour before inoculation of the virus. The progress of infection was assessed by monitoring GFP fluorescence in cells at 24 and 48 h, and virus production was quantified after 48 h by endpoint titration of virions onto Vero cells. This latter data is shown as an average of the total viral titer plotted on a logarithmic scale and, as this compresses the magnitude of differences between data points, also as a direct comparison of the treatments to the control in each experiment by conversion to a ratio that is plotted on a linear scale. Although the interferon treatment initially appeared more effective, as gauged by the levels of the fluorescence reporter after 24 h (Fig. 8F,G), this did not translate to a proportionate reduction in virus production after 48 h (Fig. 8H). While treatment with the cytokine halved virus production, treatment with JNB-hu20-051A demonstrated a 90% reduction in HSV-1 replication (Fig. 8H). Accordingly, despite only detecting a modest reduction in the expression of the reporter gene in infected cells, the JNB-hu20-051A prodrug had a greater impact on virus replication compared to the antiviral cytokine. This disparity reflects the disconnect between effects on protein translation and nucleic acid replication and appears to distinguish separate modes of action, with the interferon primarily delaying the initiation of infection while repressing the activity of the PPP reduces the replicative capacity of the virus. This curtailment of virus replication provides a proof-of-concept for a first-of-its-kind treatment to limit infection.

## Discussion

Infection imposes a significant burden on the host metabolism with high rates of nucleic acid synthesis required for viruses to be able to reproduce to high titers. This requires abundant ribose

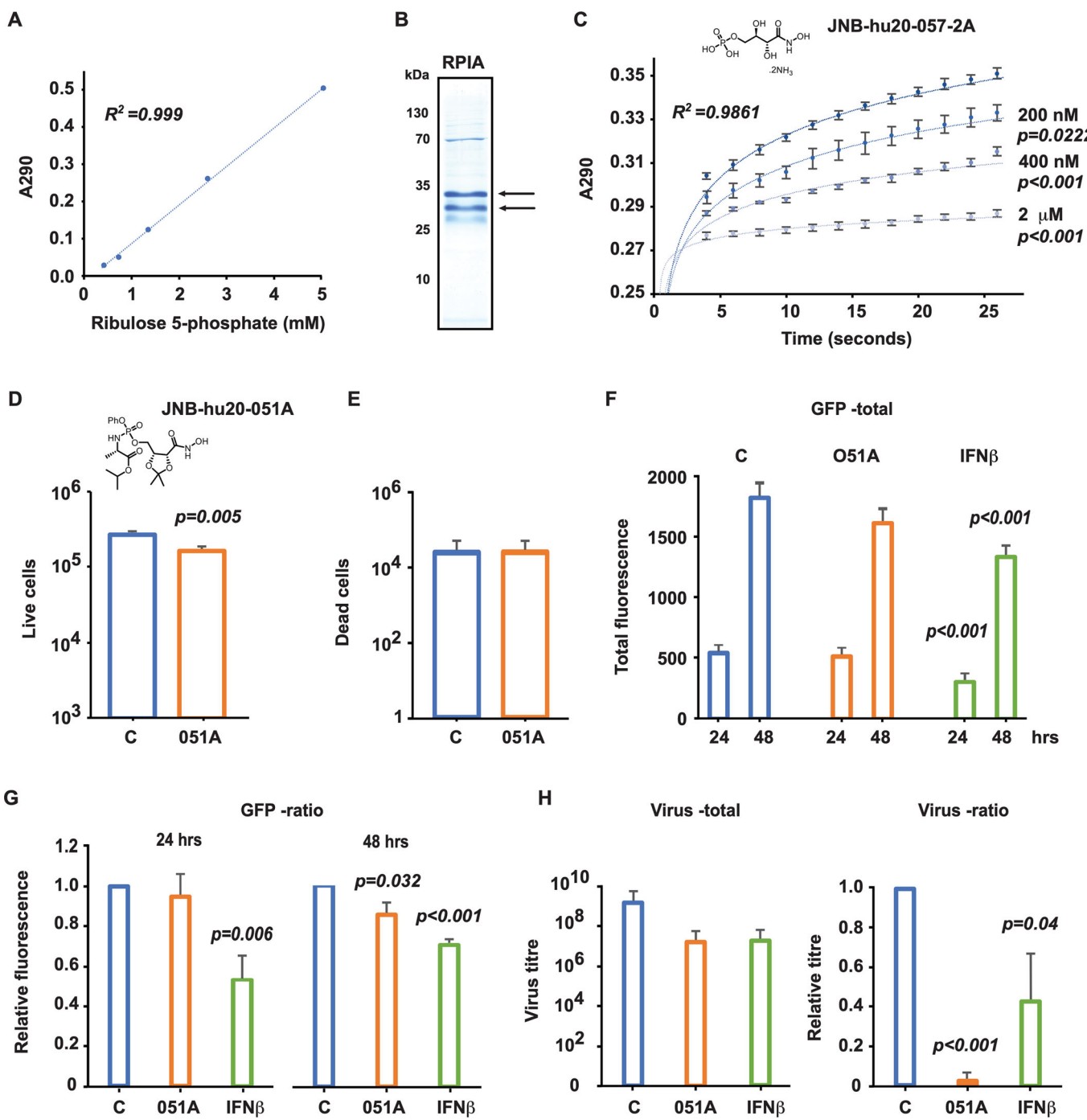

**Figure 8. Pharmacological targeting of the PPP as an antiviral.**

(**A**) A standard curve of ribulose 5-phosphate. (**B**) Coomassie-stained SDS-PAGE gel visualizing a recombinant RPIA preparation with arrows indicating different isoforms. (**C**) Plots demonstrating the effect of the indicated quantities of the inhibitor, JNB-hu20-057-2A (skeletal chemical structure inset), on RPIA activity. The P values measure the likely equivalence of the slopes (t-statistic) of linear trend lines fitted to the data ($n = 3$). The goodness-of-fit of the trend lines ($R^2$) is indicated for the control. (**D, E**) Measures of the effect of the prodrug (200 μM) JNB-hu20-051A (skeletal chemical structure inset) on (**D**) MEF growth after 48 h and (**E**) viability after 24 h as assessed with a vital stain ($n = 3$). P values were calculated by the student's t-test. (**F–H**) Plots showing the effect of the prodrug (200 μM) or IFNβ (100 iu/mL) compared to the control solvent ($H_2O$) on infection with a recombinant HSV-1-GFP strain. Infection is monitored by measuring the viral fluorescence reporter as (**F**) the total level of GFP in infected cells or (**G**) the fluorescence relative to the control at 24 ($n = 4$) and 48 ($n = 6$) hours. The data show the $\bar{x} \pm \sigma$. (**H**) Plots quantifying virus production after 48 h by endpoint dilution assay. The total virus titer is shown on the left as an average of all experiments plotted on a logarithmic scale and then replotted on the right as a ratio of the virus titer from each treatment compared to the control in separate experiments that are then averaged and shown on a linear y-axis ($n = 5$). The data shows the $\bar{x} \pm$ SEM. P values were calculated by unpaired comparisons of the treatments to the solvent calculated by the student's t-test. Figures are derived from Source Data file 8. Source data are available online for this figure.

5-phosphate, effectively tethering viral replication to the availability of this metabolite. Appropriately then, the antiviral response appears to have developed the capacity to limit the production of this critical metabolite.

Ribose 5-phosphate levels are primarily affected by de novo synthesis. Although the metabolite is also generated by recycling through nucleotide scavenging pathways, the salvage enzymes have a prerequisite for the metabolite (in the form of phosphoribosyl pyrophosphate) and so are controlled by new synthesis. Besides the ribose 5-phosphate isomerase, there are two other sources of de novo synthesis via alternative rearrangement of sugars by the transketolase within the PPP and by phosphorylation of dietary ribose by a ribokinase. The extent of the contribution of ribokinase and transketolase appears modest and is insufficient to fully compensate for isomerase activity (Brooks et al, 2018; Huck et al, 2004; van der Knaap et al, 1999). Ribose supplementation had very modest impacts in our experiments, and findings by others suggest that the activity of the ribokinase is limited to specific tissues (Clark et al, 2014). Conversion of sugar by TKT in the nonoxidative phase of the PPP proceeds through multiple steps which, because they are in equilibrium, convert only a fraction of the carbon to ribose 5-phosphate. However, all carbon entering the oxidative phase is converted to either ribose or xylulose 5-phosphate and carbon dioxide. The reversible reaction catalyzed by TKT also means that it consumes the pentose, and patients with impaired TKT activity show heightened levels of ribose 5-phosphate (Boyle et al, 2016; Li et al, 2019). Ribose 5-phosphate levels are also increased by the induction of the paralogous TKT-like 1 (TKTL1) or its retro-transposed gene product TKTL2, which associates with TKT to decrease the heterodimer's affinity for ribose compared to xylose 5-phosphate (Coy et al, 2005). Accordingly, TKT may only positively contribute to ribose 5-phosphate levels when RPIA activity is impaired (Bojkova et al, 2021). Dependence on the isomerase is further supported by its strong conservation across all species (Miosga and Zimmermann, 1996). Therefore, control of the activity of the isomerase by PKR appears as the most effective means to limit ribose 5-phosphate production.

Separate oxidative and nonoxidative phases of the PPP produce NADPH as a reducing agent for anabolism and redox control, and ribose 5-phosphate for nucleotide synthesis. Suppression of the activity of the isomerase in the nonoxidative phase preserves NADPH production in the decoupled oxidative phase of the pathway. Independent analysis by, for instance, genetic studies in drosophila confirms that NADPH production is dissociated from the activity of the isomerase (Wang et al, 2012). This is important for cell homeostasis and means that PKR's suppression of the isomerase doesn't compromise critical pathogen responses, such as the antimicrobial respiratory burst, but will disproportionately impact viruses during infection because of their asymmetric demand for the metabolite compared to the host cell.

This dependence on ribose 5-phosphate would appear to establish a strong selective pressure for a virus to secure the metabolite. Following this expectation, the genomes of bacterio-phages have been recognized to encode ribose 5-phosphate isomerases (Thompson et al, 2011). However, to our knowledge, no mammalian virus has been described that encodes the capacity to independently produce this pentose. The apparent absence of this capability in metazoan viruses may be due to reduced competition for the metabolite compared to highly replicative prokaryotic hosts. The processivity of prokaryotic polymerases approaches those of viruses, while the activity of the eukaryotic enzymes is more than tenfold lower commensurate with the reduced cell replication rate. Also, competition for ribose 5-phosphate can be decreased in mammals during infection by promoting the activity of the PPP and by curtailing the host's use of the metabolite. An occurrence of the latter is the capacity of viruses to silence ribosomal RNA biogenesis (Rawlinson et al, 2018; Selinger et al, 2019), thereby curtailing the host's major use of nucleic acids. Numerous mechanisms have been reported by which viral infection increases the flux of glucose into the PPP to increase the level of ribose 5-phosphate (Abrantes et al, 2012; Bae et al, 2017; Liu et al, 2015; Thaker et al, 2019; Vastag et al, 2011). Moreover, all viruses have developed adaptations to subvert PKR activity to preserve ribose 5-phosphate production. Despite this, virus replication rates in vivo are usually considerably below their optimum and are linearly related to the concentration of viral RNA, suggesting viruses cannot circumvent this limitation.

Recognition of the metabolic demands of viruses has propelled efforts to try to suppress metabolic activity to limit infection. Foundational in these attempts is the use of non-metabolizable glucose analogs, such as 2-deoxy-D-glucose (Kilbourne, 1959). However, outright inhibition of all glycolysis appears intolerable (Kern et al, 1982; Wang et al, 2016a). Our findings identify a more specific metabolic intervention. By limiting the rate of nucleic acid synthesis, the suppression of isomerase activity will effectively restrict virus replication. The generic requirement for the metabolite makes this a broad-spectrum response. To escape this limitation metazoan viruses would have to acquire a fundamental metabolic capability. This represents a higher genetic barrier to the development of resistance than a mutation that alters the binding of an antibody or an antiviral molecule to a specific viral target. Resistance has developed rapidly to most virus-directed treatments upon their introduction to the clinic. The dependence on ribose 5-phosphate means that control of the isomerase is likely to constitute an enduring constraint to viral infection. Here we establish a proof-of-concept for targeting the PPP to limit infection. Although the molecule we tested has low potency, its successful suppression of viral replication identifies the isomerase as a compelling antiviral target with advantages over related strategies (Bojkova et al, 2021). Our approach could satisfy a clear unmet medical need for pharmaceutical interventions to mitigate patho-genesis from current and newly emergent viral pathogens.

The recognition of the pre-existing immune control by PKR with the identification of survivors with genetic defects in *RPIA* supports tolerance for short-term pharmaceutical targeting of the activity of the isomerase during the infection period (Huck et al, 2004; van der Knaap et al, 1999). As ribose 5-phosphate constitutes several important biomolecules, this response is predicted to have wider significance. Tissues with the greatest demand for ribose 5-phosphate would appear the most exposed to the consequences of inhibiting the isomerase. This might be inferred from the expression level of *RPIA*, which is the highest in immune system organs such as lymph nodes and is most pronounced in the bone marrow (Fagerberg et al, 2014). This could foreshadow that suppressing ribose 5-phosphate production may be immune modulatory by either suppressing the expansion of lympho-cytes during the immune response or, conversely, by promoting immune cell activity by reducing the level of purines available to bind to inhibitory receptors.

Patients with a deficiency of RPIA presented with leukoencephalopathy and peripheral neuropathy (Huck et al, 2004). This is intriguing as, superficially, this phenotype appears in patients on extended treatment with type I interferons (Kleinschmidt-DeMasters and Tyler, 2005; Lehmann et al, 2015; Metzler et al, 2005), or with genetic defects that cause type I interferonopathies (Rice et al, 2013) and with viral-dependent induction of interferon in the brain (Cortese et al, 2021; Lee et al, 2018). This phenotype is also recorded experimentally in mice with defects in interferon-dependent signaling (Aicardi and Goutieres, 1984; Kettwig et al, 2021). Our findings may identify that this pathophysiology could be a consequence of persistent suppression of RPIA activity by interferon-induced PKR (Thomis et al, 1992). This could identify the benefit of PKR inhibitors in patients with interferonopathies or on extended interferon treatment.

Our findings recognize a feedback regulatory loop that had been apparent for the other EIF2α kinases but which was less evident for PKR. The activity of the heme-regulated inhibitor kinase is regulated by protoheme, which accrues from repressed hemoglobin synthesis. Similarly, the PKR-like endoplasmic reticulum kinase is regulated by protein chaperones which accumulate after inhibiting the synthesis of client proteins. Like PKR, GCN2 activity is induced by an activating RNA ligand. Low levels of amino acids lead to increasing amounts of uncharged tRNA, which directly bind to GCN2. This activating tRNA decreases with increasing levels of amino acids following the inhibition of protein translation. Although EIF2α phosphorylation sequesters RNA into stress granules and induces autophagy, this does not rapidly reduce the levels of PKR's activating ligand. However, regulation of the isomerase would dampen the ongoing production of RNA to relieve chronic PKR activity. As limiting RNA would also appear likely to affect the levels of the activating ligand for GCN2, it would be interesting to test if RPIA is also phosphorylated by GCN2. This would establish control of the isomerase as part of the prototypical response by this kinase family.

Despite differences between prokaryotes and eukaryotes, there are intriguing parallels in the control of the initiation of protein translation in response to environmental stress. Homologous to GCN2 in eukaryotes, the bacterial GTP pyrophosphokinase (RelA) senses amino acid starvation through the binding of uncharged tRNA. Activated RelA then hyper-phosphorylates guanosine to produce the nucleotide messenger alarmone that establishes the stringent response. Alarmone remodels the metabolism by binding to various enzymes, including the translational initiation factor 2 to suppress protein synthesis and the ribose isomerase to inhibit nucleotide synthesis. As ribose 5-phosphate is at the center of many of the metabolic switches, it has been proposed as a bridging molecule that synchronizes adaptation to stress (Grucela et al, 2023; Liu et al, 2020). Although metazoans do not appear to produce significant amounts of alarmone, they employ other nucleotide phosphates as intracellular signaling molecules and retain vestiges of the prokaryotic stringent response (Ito et al, 2020). Interestingly, disruption of the metazoan alarmone hydrolase gene (*Metazoan SpoT Homolog 1 (MESH1)*) triggers a stress response that suppresses the activity of the PPP and is accompanied by the accumulation of polyols and induces the EIF2α kinase response (Ito et al, 2020; Lin et al, 2021; Sun et al, 2010; Witowski et al, 2015). This entanglement with the common elements and similar outcomes leads us to speculate that the primaeval stringent response has been supplanted in eukaryotes by the EIF2α kinases, with their phosphorylation control extending to the phosphoribosyl isomerase.

In summary, we have identified a previously unknown PKR substrate that constitutes an unrecognized antiviral activity. This instigates an immune-mediated metabolic adaptation to counter viral infection through phospho-control of RPIA. Although viruses have developed adaptations to subvert PKR activity, they remain dependent on the isomerase for ribose 5-phosphate and so this identifies a vulnerability that may be exploited to try to limit pathogenesis from infectious viruses.

## Data availability

All data were available in the main text or the supplementary materials or are available from public repositories. Metabolic data are available at the Metabolomics Workbench, with the ID ST002412. Phosphorylation data are available via the PRIDE repository, with the ID PXD036779.

The source data of this paper are collected in the following database record: biostudies:S-SCDT-10_1038-S44318-024-00100-w.

## Peer review information

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

## Acknowledgements

The authors are grateful to the National BioResource Project, National Institute of Genetics (Japan) for the provision of libraries of mobile genetic elements of *E. coli* metabolic factors; John Boyce, Microbiology Dept, Monash University for providing the *pWKS130* vector, and Antony Matthews, working with Paul Hertzog, for providing the recombinant murine IFNβ. The authors acknowledge the use of the services and facilities of Micromon Genomics and Bio-platforms Australia, National Collaborative Research Infrastructure Strategy-enabled infrastructure located at the Monash Proteomics and Metabolomics Facility, as well as the Monash Flowcore at the Monash Health Translational Precinct, Translational Research Facility. Additional gratitude is due to the NIH Common Fund's National Metabolomics Data Repository and the ProteomeXchange

Consortium website for hosting data. AJS was supported by an NHMRC grant (1143839). DW was supported by an NHMRC grant (1043398) and a philanthropic Perpetual Trustees grant (CF07/2408) awarded to AJS. PB was supported by a philanthropic Perpetual IMPACT grant awarded to AJS. JB and LH were supported by the New Zealand Ministry of Business Innovation & Employment (Endeavour Fund UOOX1904).

## Author contributions

**Pushpak Bhattacharjee**: Data curation; Investigation; Methodology. **Die wang**: Investigation; Methodology. **Dovile Anderson**: Formal analysis; Investigation; Methodology. **Joshua, N Buckler**: Investigation; Methodology. **Eveline, de Geus**: Investigation; Methodology. **Feng Alex Yan**: Software; Formal analysis. **Galina Polekhina**: Software; Formal analysis. **Ralf Schittenhelm**: Resources; Data curation; Software; Formal analysis; Supervision; Investigation; Methodology. **Darren, J Creek**: Resources; Data curation; Software; Formal analysis; Supervision; Investigation; Methodology. **Lawrence, D Harris**: Resources; Formal analysis; Supervision; Funding acquisition; Investigation; Methodology. **Anthony, J Sadler**: Conceptualization; Data curation; Formal analysis; Supervision; Funding acquisition; Investigation; Methodology; Project administration.

Source data underlying figure panels in this paper may have individual authorship assigned. Where available, figure panel/source data authorship is listed in the following database record: biostudies:S-SCDT-10_1038-S44318-024-00100-w.

## Disclosure and competing interests statement

The authors declare no competing interests.

