## [Peer Review File · The EMBO Journal]

The immune response to RNA suppresses nucleic acid synthesis by limiting ribose 5-phosphate

Pushpak Bhattacharjee, Die Wang, Dovile Anderson, Joshua N. Buckler, Eveline de Geus, Feng Alex Yan, Galina Polekhina, Ralf Schittenhelm, Darren J. Creek, Lawrence D. Harris, Anthony J. Sadler

Corresponding author: Anthony Sadler (anthony.sadler@hudson.org.au)

Review Timeline:

Submission Date:	6th Sep 23
Editorial Decision:	10th Nov 23
Revision Received:	21st Dec 23
Editorial Decision:	7th Feb 24
Revision Received:	29th Feb 24
Accepted:	19th Mar 24

Editor: Daniel Klimmeck

Transaction Report:

Dear Dr Sadler,

Thank you again for the submission of your manuscript (EMBOJ-2023-115524) to The EMBO Journal and in addition providing us with a preliminary revision plan. As mentioned earlier, your study was assessed by three reviewers with expertise in cellular metabolism, RNA biology and antiviral immunity, whose comments are enclosed below.

As you will see from their comments, the referees acknowledge the analysis and potential interest and value of your findings. However, they also express major concerns i.p. regarding the extent of causal support for the claimed moonlighting role of PKR in R5P generation and redox metabolism upon viral RNA exposure. They also have important concerns on overall robustness of the results and request s complementary experiments to exclude potential confounding factors. Further, the reviewers raise a number of points related to the overall structure of the manuscript and presentation of the findings, additional controls required, improved methods annotation and data processing and overall discussion of related literature, that would need to be conclusively addressed to achieve the level of robustness and clarity needed for The EMBO Journal.

Given the overall interest stated and broader angle of your findings, we are able to invite you to revise your manuscript experimentally to address the referees' comments, along the lines sketched in your outline. I need to stress though that we do require strong support from the referees on a revised version of the study in order to move on to publication of the work. As to the open outcome of the major revisional work required, I suggest keeping EMBO Reports in mind for this study as an alternative venue.

Please feel free to contact me if you have any questions or need further input on the referee comments.

When submitting your revised manuscript, please carefully review the instructions below.

Please feel free to approach me any time should you have additional questions related to this.

Thank you for the opportunity to consider your work for publication.

I look forward to your revision.

Best regards,

Daniel Klimmeck

Daniel Klimmeck, PhD
Senior Editor
The EMBO Journal

Instruction for the preparation of your revised manuscript:

- 1) a .docx formatted version of the manuscript text (including legends for main figures, EV figures and tables). Please make sure that the changes are highlighted to be clearly visible.
- 2) individual production quality figure files as .eps, .tif, .jpg (one file per figure).
- 3) a .docx formatted letter INCLUDING the reviewers' reports and your detailed point-by-point response to their comments. As part of the EMBO Press transparent editorial process, the point-by-point response is part of the Review Process File (RPF), which will be published alongside your paper.
- 4) a complete author checklist, which you can download from our author guidelines ([https://wol-prod-cdn.literatumonline.com/pb-assets/embo-site/Author Checklist%20-%20EMBO%20J-1561436015657.xlsx](https://wol-prod-cdn.literatumonline.com/pb-assets/embo-site/Author%20Checklist%20-%20EMBO%20J-1561436015657.xlsx)). Please insert information in the checklist that is also reflected in the manuscript. The completed author checklist will also be part of the RPF.

6) It is mandatory to include a 'Data Availability' section after the Materials and Methods. Before submitting your revision, primary datasets produced in this study need to be deposited in an appropriate public database, and the accession numbers and database listed under 'Data Availability'. Please remember to provide a reviewer password if the datasets are not yet public (see <https://www.embopress.org/page/journal/14602075/authorguide#datadeposition>).

7) Our journal encourages inclusion of *data citations in the reference list* to directly cite datasets that were re-used and obtained from public databases. Data citations in the article text are distinct from normal bibliographical citations and should directly link to the database records from which the data can be accessed. In the main text, data citations are formatted as follows: "Data ref: Smith et al, 2001" or "Data ref: NCBI Sequence Read Archive PRJNA342805, 2017". In the Reference list, data citations must be labeled with "[DATASET]". A data reference must provide the database name, accession number/identifiers and a resolvable link to the landing page from which the data can be accessed at the end of the reference. Further instructions are available at .

8) At EMBO Press we ask authors to provide source data for the main and EV figures. Our source data coordinator will contact you to discuss which figure panels we would need source data for and will also provide you with helpful tips on how to upload and organize the files.

Numerical data can be provided as individual .xls or .csv files (including a tab describing the data). For 'blots' or microscopy, uncropped images should be submitted (using a zip archive or a single pdf per main figure if multiple images need to be supplied for one panel). Additional information on source data and instruction on how to label the files are available at .

9) We replaced Supplementary Information with Expanded View (EV) Figures and Tables that are collapsible/expandable online (see examples in <https://www.embopress.org/doi/10.15252/embj.201695874>). A maximum of 5 EV Figures can be typeset. EV Figures should be cited as 'Figure EV1, Figure EV2" etc. in the text and their respective legends should be included in the main text after the legends of regular figures.

11) For data quantification: please specify the name of the statistical test used to generate error bars and P values, the number (n) of independent experiments (specify technical or biological replicates) underlying each data point and the test used to calculate p-values in each figure legend. The figure legends should contain a basic description of n, P and the test applied. Graphs must include a description of the bars and the error bars (s.d., s.e.m.).

We realize that it is difficult to revise to a specific deadline. In the interest of protecting the conceptual advance provided by the work, we recommend a revision within 3 months (14th Jan 2024). Please discuss the revision progress ahead of this time with the editor if you require more time to complete the revisions.

Referee #1:

In this report, Bhattacharjee et al. describe how Protein Kinase RNA-activated (PKR) is involved in altering glucose breakdown within the pentose phosphate pathway by inhibiting the conversion of ribulose 5-phosphate to ribose 5-phosphate. The authors demonstrate that Ribose 5-Phosphate Isomerase A (RPIA) can be phosphorylated by PKR, and this phosphorylation leads to the inhibition of the pentose phosphate pathway (PPP), which exhibits antiviral properties.

General comments:

Many experiments presented in this paper lack essential controls, undermining the support for the authors' conclusions. Importantly, despite several suggestive results, the authors persist with their interpretations without providing additional evidence to substantiate their conclusions or the proposed causal links between observations. Additionally, the concept of modifying the ribose 5-phosphate production pathway to control viral replication is not novel. The relevant literature on PPP inhibition and its consequences on viral infection is contradictory, as evidenced by studies on Zika Virus and early HIV1 infection, which show no direct effect and essentiality, respectively.

Although WT and PKR knockout cells are used in most experiments, there is no prior characterization of their biochemical and immunological status towards Poly:I:C exposure documented. This includes the state of eIF2A phosphorylation, protein synthesis levels, and cytokine production, including type-I IFN. The absence of evidence regarding a decrease in eIF2A-P levels and the potential lack of translational arrest in PKR-KO cells raises questions about how these may result into the observed metabolic changes in Figure 1. Higher glucose consumption due to increased protein synthesis could significantly modify metabolism. Experiments should have been conducted in WT cells using ISRIB and/or a PKR inhibitor to confirm or refute the results obtained with PKR KO cells. Importantly, inhibiting protein synthesis using an EIF4G1 inhibitor could mimic EIF2AK activation in PKR KO cells and confirm or refute the direct role of PKR in interfering with the PPP. Additionally, different autocrine responses to IFN-B in WT or EIF2KA cells upon Poly-I:C delivery could also impact glucose metabolism. Experiments in IFNAR KO cells or in the absence of effective IFN-B activity should be added.

The method and delivery of poly I:C are poorly described. It's important to consider that parallel activation of TLR3 could trigger IFN in macrophages independently of intracellular delivery, potentially altering cell reactivity.

The use of a prokaryotic assay to monitor eukaryotic kinase activity through overexpression is problematic. Yeast and yeast two-hybrid systems might have been more suitable. The phosphatase used to inhibit PKR activity in *E. coli* is not identified. The experiments lack essential controls, including the expression of other eIF2A kinases besides PKR and/or kinase domains, to demonstrate the specificity of the observation and at least rule out off-target effects.

In Figure 4, contrary to the text description, protein levels in the blot appear different in WT and eIF2K ^{-/-} extracts for G6PD, PDGLS, PDG, RPIA, TKT, and G6PDH. These experiments need to be repeated and quantified more accurately to support the authors' claims. This again relates to the status of protein synthesis in the cells (not documented) and its overall effect on metabolism.

The efficiency of silencing by shRNA is not documented, and the rationale for changing cell types and gene inactivation technique at this stage of the manuscript is unclear. Importantly, shRNA can modify the IFN response by triggering innate receptors and could have extremely different impact than gene deletion. Again the impact on eIF2A phosphorylation, protein synthesis levels and cytokine production during silencing is not documented.

Comparison of cell lines used in different assays by RNAseq would provide a clearer view of the differences induced by PKR deficiency and support the authors' conclusions.

In Figure 4L, the authors use several useful controls, but they are not used in the PKR KO line, and are introduced late in the paper, without any demonstration of successful ectopic expression shown. ATF4 silencing/downregulation (again not documented) seems capable of counteracting PKR inactivation, suggesting the existence of a relationship established between the PPP cycle and the ISR. However, the authors quickly disregard this observation, that nevertheless indicates a potential role for translation control (ATF4 is normally translated upon protein synthesis arrest) in controlling glucose metabolism, despite the lack of documentation of ATF4 expression and translation in their experimental setting.

In Figure 5, the experiments should also be performed with a specific PKR inhibitor to exclude any contaminating kinase activity coming from the purification procedure. It is important to clarify how PKR is activated in the assay. Additionally, controlling the specificity of the activity with another eIF2aK should be done.

In Figure 6, the quality of the microscopy is poor, lacking quantification and secondary labeling for G3BP1-positive stress granules known to contain PKR upon eIF2 phosphorylation. The figure does not adequately support the conclusions in the text.

The effect of the JNB inhibitor on HSV1 replication is perplexing, showing a block on the viral GFP reporter and an effect on the relative replication titer. A careful study of the inhibitor's impact should be performed in a separate manuscript, independently of the PKR story, as PPP modulation is currently proposed as an anti-viral target.

The authors have not directly demonstrated that phosphorylation of RPIA by PKR has an effect on viral expression. Additionally, the conclusion on the effect of RPIA phosphorylation (or lack thereof) by PKR in *E. coli* is confusing, suggesting again an off-target effect of the kinase on bacterial growth.

Minor: Figure numbers should be indicated on the figures, and the manuscript's writing could be considerably simplified.

Referee #2:

In the manuscript titled "The immune response to RNA suppresses nucleic acid synthesis by inhibiting ribose 5-phosphate production", the authors shed light on the metabolic impact of the antiviral effector Protein Kinase RNA-Activated (PKR), a previously under-researched area. The paper illustrates PKR's role in thwarting viral proliferation by reducing pentose precursor molecules, essential for viral nucleotide production.

The study reveals that PKR deletion significantly affects macrophage metabolism during RNA challenge. This includes increased levels of PPP intermediates but decreased levels of xylose-5-phosphate.

The authors employ an innovative bacterial assay to explore potential metabolic enzyme phosphorylation by PKR, focusing on phosphoribosyl isomerase (RpiA) among others.

The research also discusses PKR-induced suppression of RpiA using, inter alia, phosphotransfer assays. Subsequent experiments spotlight the structural context of phospho residues crucial for enzyme activity. Moreover, an inhibitor of RpiA hints at its potential importance in antiviral activity.

Major points

The paper uses metabolite abundance profiles as a surrogate for cellular flux. However, a decrease in metabolites does not necessarily mean that there is a reduction in pathway activity, as is suggested at multiple occasions throughout the manuscript. Additional data, at minimum isotopologue analysis, would be required to make any statements on flux.

There is no apparent quenching step for metabolomics analysis, but cells seem to be trypsinized, suspended in PBS, centrifuged, and then further processed at 4°C. PPP intermediates have a very fast turnover, and the described procedure will likely not conserve the actual metabolic state. I thus have doubts about the relevance of these specific findings, and the authors should at least disclaim that this likely affects their data.

The most striking difference between WT and *Eif2ak2*^{-/-} seems to be that there are severely reduced levels of glucose-6-phosphate. This would lead to reduced PPP pathway intermediates. Based on this, I would suspect that the major effect of the deletion is due to increased conversion of glucose to glucose-6-phosphate. The increase in xylose 5-phosphate seems a diversion from this pattern, but may not be the underlying effect. I think the authors should comment on this.

I am a bit puzzled by the results from the bacterial assays probing for potential phosphorylation targets by co-expressing substrates of PKR with PKR. The majority of metabolic enzymes seem to rescue the defect, and would thus be considered substrates of PKR. Is this expected? Further, if all these enzymes are targets and thus limit growth, how can expression of a single enzyme rescue it? Is there such a strong degree of phosphorylation capacity of PKR then used on the overexpressed enzyme? Please clarify.

It was not apparent to me how the data was normalized in Figure 8 G and H (right panel). Is there an initial timepoint that is not shown? As there are such striking differences between relative and absolute data, it would be helpful to understand (e.g. in the figure legend) how this is normalized.

How do the authors explain the strong difference between the GFP-based quantification of viral load, and the viral titer?

The antiviral strategy targeting mammalian RpiA seems to be a cytostatic mode of action for the host cell. Is this expected to

have strong side-effects in proliferative cell types, that would themselves rely on ribose-5-phosphate supply?

Overall understandability and presentation of data could be improved. I believe the manuscript could be more accessible to a reader with a bit more concise background and illustrations.

For instance, in the figures, a small outline of the RNA-response pathway, as well as the respective metabolic pathways and signaling cascades studied would be helpful. Similarly, figures such as Figure 2C use a lot of space for limited information - surely there can be a better way to illustrate this, with the original images moved to the supplement

The title is not clear - nucleic synthesis where? In which cells? Immune response against what?

The abstract is not very concise, and remains vague, using phrases such as

The pathway "recycles" carbon - what should that mean in this context?

"Changes to the level of metabolic intermediates within this pathway identify an effect on ..." - the intermediates surely don't identify an effect?

Again, there is information in the abstract where this is happening - in immune cells? Epithelial cells? In vivo? In vitro? The abstract should make clear what was studied.

The introduction lacks crucial references. Many statements throughout the introduction are not supported by any references at all, including very specific findings that are certainly not common knowledge.

Minor

- the high levels of X5P and R5P (Text for Figure 1D) in WT strain are striking here which was mentioned in the text. It might be helpful to include the pooled results of measurements of R5P and Ru5P and add it to Figure 1D.

- you could include an additional figure in supplement which gives more transparency of all measured products of X5P and R5P -> to emphasize that all products show lower amounts in WT cells (not only adenosine)

- The metabolite X5P seems accumulated in WT cells

-> does X5P have other roles, for example in signaling? Could there be the option that its accumulation might display an advantage for antiviral response?

-> it is for example known that Xu5P is a signaling compound and involved in activation of

PPases controlling fat synthesis and glucose metabolism (PMID: 7929321, PMID: 12684532)

- Did you ever test also macrophages derived from liver cells beside the tested spleen and bone-marrow derived macrophages? The ribokinase activity seems for example to be tissue specific, so would be interesting to see if the mechanism is still similar in this liver tissue

- Figure 3: suggestion to use different colors as it's a bit confusing to have the same colors orange and blue in a slightly different context (Figure 3B)

Referee #3:

This manuscript describes a previously unidentified mechanism by which PKR modulates the response to dsRNA, independently of eIF2 phosphorylation and translation arrest. The major conclusion of this work is that PKR modulates glucose metabolism via suppression of phosphoribosyl isomerase (RPIA) activity, subsequently reducing the rate of ribonucleotide synthesis via the pentose phosphate pathway (PPP). This conclusion is supported by several experimental results. First, metabolomics analysis showed that many PPP metabolites are lower in WT macrophages treated with the viral RNA mimetic poly(I:C), compared to PKR-depleted macrophages. Kinase assays demonstrated that RPIA is a PKR substrate in vitro. The physiological relevance of the kinase assay is supported by the colocalization of RPIA and kinase-active PKR in assays performed in cells. Finally, HSV replication is reduced in cells treated with an RPIA inhibitor in comparison to untreated cells, implicating RPIA as a therapeutic target in the context of viral infection. The identification of RPIA as a PKR substrate is significant, as it provides a broader understanding of how PKR modulates cell physiology and may also have implications for understanding the role of PKR in metabolic disease. I found the work of interest and following revision to address the comments below, it might be appropriate for publication in EMBO.

Comments:

1) The major conclusion of this work is that RPIA is phosphorylated by PKR upon dsRNA activation in mammalian cells. This is strongly supported by evidence that PKR can phosphorylate RPIA in vitro. It also needs to be strongly supported by evidence from mammalian cells. The key experiment is shown in Figure 5D. However, this figure is very difficult to interpret since there are additional smudges in the RPIA lane, and the lanes are arranged in a way that makes it difficult to compare the relative

position of different protein species. It is important to improve the clarity of this experiment, or perform a related experiment, convincingly demonstrating PKR phosphorylation of RPIA in mammalian cells in response to dsRNA.

2) As a reader, I found this manuscript difficult to follow and discern the logic and key results. To improve the manuscript, I suggest the authors: a) Focus on the important results and experiments and potentially remove or move to the supplemental experiments that are of lower priority. b) Wherever possible, be sure the logic and the critical results are emphasized. Improving the logical presentation of the work will allow this manuscript to be appreciated by a wide audience.

3) The authors should be aware that other groups have targeted the PPP as an antiviral strategy. This does not detract from the importance of this manuscript, but the authors should discuss those previous manuscripts (e.g. Bojkova et al., 2021, Metabolites).

Authors responses.

Referee #3:

1) The major conclusion of this work is that RPIA is phosphorylated by PKR upon dsRNA activation in mammalian cells. This is strongly supported by evidence that PKR can phosphorylate RPIA in vitro. It also needs to be strongly supported by evidence from mammalian cells. The key experiment is shown in Figure 5D. However, this figure is very difficult to interpret since there are additional smudges in the RPIA lane, and the lanes are arranged in a way that makes it difficult to compare the relative position of different protein species. It is important to improve the clarity of this experiment, or perform a related experiment, convincingly demonstrating PKR phosphorylation of RPIA in mammalian cells in response to dsRNA.

Phosphorylation of RPIA by PKR is also demonstrated in a second mammalian cell line by mass spectrometric identification of phosphoresidues (Table 1, PRIDE repository ID PXD036779 and Fig 6). Some additional data is presented as Supplementary data to demonstrate PKR activity in this experiment (SF8B).

The blots in Fig 5D have been edited to change their size and arrangement to display the protein species better.

2) As a reader, I found this manuscript difficult to follow and discern the logic and key results. To improve the manuscript, I suggest the authors: a) Focus on the important results and experiments and potentially remove or move to the supplemental experiments that are of lower priority. b) Wherever possible, be sure the logic and the critical results are emphasized. Improving the logical presentation of the work will allow this manuscript to be appreciated by a wide audience.

We altered the order in which the results were presented. Experiments in macrophages are shown first with experiments in bacteria following. We moved some of the data shown in Fig 2C to supplementary material (SF2) to focus on the PPP enzymes and the related isomerase (RpiB), as was suggested. Data are now also grouped differently with metabolic data Fig 1, analysis of ribose 5-phosphate products Fig 2 and the investigation of PKR signalling in macrophages followed by the experiments investigating PKR activity in bacteria in Fig 3. The remaining order of data is as before, although the order of panels in Fig 5 is changed to try to make this more intuitive. As mentioned above, the immunoblots in Fig 5 are also altered to improve interpretation. We changed the presentation of some data so that it consistently shows measures relative to the WT/shCONT cells. The colour coding of Fig 4C & D and Fig 7E was altered, as suggested by another Reviewer. We also moved a plot of the levels of NADPH/NADP⁺ from supplementary data into Fig 2 to support the explanation that PKR activity controls the nonoxidative and not the oxidative phase. The order of the Supplementary figures has been changed accordingly and some additional data has been added to the appendix to support the findings shown in the main text (as listed in

this response). We have made the abstract more concise and a graphical abstract has also been added to summarise the findings.

3) The authors should be aware that other groups have targeted the PPP as an antiviral strategy. This does not detract from the importance of this manuscript, but the authors should discuss those previous manuscripts (e.g. Bojkova et al., 2021, Metabolites).

Although the concept proposed by Bojkova et al. supports our approach to suppress R5P production, we hadn't cited this report as we had not found it convincing. Conversion of carbon to R5P via the nonoxidative phase of the PPP proceeds through multiple steps (S7P+G3P, E4P+F6P, E4P+X5P, R5P+X5P). As all products are at equilibrium, only a fraction of the available carbon can be converted to R5P by this route. However, all the ribulose 5-phosphate produced in the oxidative phase of the PPP can be converted to either R5P or X5P making RPIA activity more consequential. Both enzymes can also consume R5P. This loss is more telling for TKT as its products are subsequently removed by TALDO1 (evident in Fig 2F), while the RPIA product is insufficient for subsequent metabolism via the PPP. Accordingly, TKT reduces R5P levels. This is evident in patients with impaired TKT activity who have heightened levels of R5P. TKT likely positively contribute to R5P levels only in instances where RPIA activity is impaired. This may have been the case in the infected cells in Bojkova's experiments through the mechanism we identify here. There are other uncertainties in this report. Two inhibitors were used to suppress SARS-Cov-2 infection. The first was the unmetabolizable 2-deoxy-d-glucose, which we had discussed as an antiviral strategy and cited the original report of this approach. The second was a thiamine antagonist. Thiamine is a cofactor for numerous enzymes involved in carbohydrate metabolism, including TKT, α -ketoglutarate dehydrogenase, pyruvate dehydrogenase, the branched-chain α -keto acid dehydrogenase and the 2-hydroxyacyl-CoA lyase. Accordingly, the drug will inhibit the PPP and multiple steps in the tricarboxylic acid cycle. There was no metabolic analysis to distinguish how the drug was working and no specific measures of the dependence for TKT for the observed effects.

Despite these misgivings, we now cite Bojkova et al as part of our discussion of alternative routes to produce R5P and antiviral strategies that target the PPP.

Referee #2:

The paper uses metabolite abundance profiles as a surrogate for cellular flux. However, a decrease in metabolites does not necessarily mean that there is a reduction in pathway activity, as is suggested at multiple occasions throughout the manuscript. Additional data, at minimum isotopologue analysis, would be required to make any statements on flux.

The measures of the levels of metabolites are not intended to measure cellular flux but to demonstrate the changed levels of different metabolites within the PPP. This identifies altered enzyme activity. Notably, all steps in the non-oxidative phase are reversible and disposed to equilibrium so the uneven distribution of the levels of metabolites within this pathway is instructive. We have tried to remove any perceived suggestion of differences in the rate of glucose oxidation.

There is no apparent quenching step for metabolomics analysis, but cells seem to be trypsinized, suspended in PBS, centrifuged, and then further processed at 4°C. PPP intermediates have a very fast turnover, and the described procedure will likely not conserve the actual metabolic state. I thus have doubts about the relevance of these specific findings, and the authors should at least disclaim that this likely affects their data.

The cells were recovered in cold PBS and, as noted, suspended in a cold solvent and processed at 4°C and in liquid nitrogen. The WT and knockout cells are processed together, thereby ensuring that the differences between them are not technical. A statement to this effect has been added to the methods.

This lability may be overstated as we conducted timed experiments with labelled glucose that showed little change to the PPP products with different processing protocols. It may require a substantive draw on a particular metabolite to see such a change.

The most striking difference between WT and Eif2ak2^{-/-} seems to be that there are severely reduced levels of glucose-6-phosphate. This would lead to reduced PPP pathway intermediates. Based on this, I would suspect that the major effect of the deletion is due to increased conversion of glucose to glucose-6-phosphate. The increase in xylose 5-phosphate seems a diversion from this pattern, but may not be the underlying effect. I think the authors should comment on this.

The different levels of glucose 6-phosphate stem from the heightened production of fructose 6-phosphate in the PPP in the absence of PKR. Notably, there is not a major difference in other intermediates of the glycolytic pathway that aren't shared with the PPP, so there is no support for an altered rate of glucose phosphorylation. The flow of carbon from the nonoxidative PPP back through the upper glycolytic pathway accounts for this normalisation of intermediates lower in the glycolytic pathway and the elevated levels of glucose 6-phosphate. We added this explanation to the text to clarify this point and altered Fig 1D to indicate this cycling of carbon. We also added two references that

report the recycling of carbon through the PPP and upper glycolytic pathways to support this explanation (PMIDs 18154684 & 26190262).

I am a bit puzzled by the results from the bacterial assays probing for potential phosphorylation targets by co-expressing substrates of PKR with PKR. The majority of metabolic enzymes seem to rescue the defect, and would thus be considered substrates of PKR. Is this expected? Further, if all these enzymes are targets and thus limit growth, how can expression of a single enzyme rescue it? Is there such a strong degree of phosphorylation capacity of PKR then used on the overexpressed enzyme? Please clarify.

RPIA is the only PKR substrate identified in these experiments. The other proteins recover growth because they aren't phosphorylated by PKR. The kinase is expressed at a high level in the bacteria and is highly processive so phosphor-control can't be overcome by increasing the expression of a substrate. The rescue is due to the restoration of the levels of ribose 5-phosphate. The enzymes in the non-oxidative phase of the pathway rescue growth by diverting carbon from the glycolytic pathway to alternatively supply R5P. This is confirmed by using inactive mutant constructs. The enzymes in the oxidative phase did not rescue growth as their activity still required the isomerase to convert the product from this phase (Ru5P).

The PKR substrate is alternatively identified, without expressing the PPP enzymes, by using single carbon sources that require isomerase activity. Importantly, overexpressing a different isomerase (RpiB) that is not a PKR substrate also rescues growth by alternatively generating the missing R5P. We have tried to clarify the description of these results.

It was not apparent to me how the data was normalized in Figure 8 G and H (right panel). Is there an initial timepoint that is not shown? As there are such striking differences between relative and absolute data, it would be helpful to understand (e.g. in the figure legend) how this is normalized.

The data in H shows the total virus titre (on the left) and the same data normalized to the level in the control (on the right). The conspicuous difference is due to the change from a logarithmic to a linear scale on the y-axis of the plot. A similar comparison is made in F and G, with the total levels of fluorescence (GFP) shown in F replotted in G as the levels relative to the control. The difference is less conspicuous as both sets of data are plotted on a linear scale.

This is described in the figure legend: "Infection is monitored by measuring the viral fluorescence reporter as **(F)** the total level of GFP in infected cells or **(G)** the fluorescence relative to the control at 24 and 48 hours. **(H)** The infection was also assessed by quantifying virus production after 48 hours by end-point dilution assay. The total virus titre is shown on the left and replotted on the right relative to the untreated control with a linear y-axis."

How do the authors explain the strong difference between the GFP-based quantification of viral load, and the viral titer?

Protein and nucleic acid production are not precisely linked. The treatment elaborates this difference by disproportionately affecting the replication of the viral genome (ribose 5-phosphate production) relative to the rate of protein translation (evident as GFP).

The antiviral strategy targeting mammalian RpiA seems to be a cytostatic mode of action for the host cell. Is this expected to have strong side-effects in proliferative cell types, that would themselves rely on ribose-5-phosphate supply?

This is the case during in vitro culture. Most cells in the body aren't highly replicative so would be less affected by a short-term suppression of RPIA activity. Having said this, the immune response requires lymphocyte replication so an induced nucleotide deficiency could suppress the T- and B-cell response. Reports suggest that the cell cycle control machinery upregulates the expression of transketolase isoforms (TKTL1 & TKTL2, while suppressing the expression of TKT itself, PMID 31175280), which could partly restore R5P in lymphocytes. Accordingly, the impact of suppressing RPIA activity isn't clear. Countering a possible suppression of immune cell expansion, the induced nucleotide deficiency should stimulate immune activity through reduced binding of purines to inhibitory receptors on immune cells. A form of this text is in the manuscript discussion.

Overall understandability and presentation of data could be improved. I believe the manuscript could be more accessible to a reader with a bit more concise background and illustrations. For instance, in the figures, a small outline of the RNA-response pathway, as well as the respective metabolic pathways and signaling cascades studied would be helpful. Similarly, figures such as Figure 2C use a lot of space for limited information - surely there can be a better way to illustrate this, with the original images moved to the supplement.

We have attempted to improve the clarity of the manuscript with the changes described above. These adopted the Reviewer's suggestion for Fig 2 in the edited manuscript. The metabolic pathways were shown diagrammatically in Fig 1 and Fig 4 (previously Fig 3). The previously established PKR response is described in the introduction. As this primarily involved one substrate (EIF2alpha) it would not seem to require a diagram. PKR signalling becomes more complicated over longer timeframes, with the induction of the integrated stress response. However, the beauty of our bacterial experiments is that they demonstrate that PKR acts directly on the isomerase, independent of the integrated response (as these are absent in E. coli).

The title is not clear - nucleic synthesis where? In which cells? Immune response against what?

The title describes the immune response as being against RNA. As PKR and RPIA are universally expressed the response is ubiquitous in cells with the activating stimuli (duplex RNA).

The abstract is not very concise, and remains vague, using phrases such as The pathway "recycles" carbon - what should that mean in this context?

We deleted 'recycle carbon' and edited the abstract to try to make it more concise.

"Changes to the level of metabolic intermediates within this pathway identify an effect on ..." - the intermediates surely don't identify an effect?

The sentence is reworded as 'Changes in the levels of metabolites between wild-type and PKR-ablated macrophages identify that the kinase controls the conversion of ribulose to ribose 5-phosphate'.

Again, there is [no?] information in the abstract where this is happening - in immune cells? Epithelial cells? In vivo? In vitro? The abstract should make clear what was studied.

The particular cell used for metabolic analysis is reported (macrophages). All cells have the capability for this response as the proteins are universally expressed.

The introduction lacks crucial references. Many statements throughout the introduction are not supported by any references at all, including very specific findings that are certainly not common knowledge.

References have been added to support the discovery of the different EIF2 α kinases, RNA-binding by PKR and induction of the kinase by IFN that we surmise were unsupported statements.

As the manuscript had considerably more references than designated by the journal we removed some citations related to possible effects of the response in immune cells from the discussion.

Minor

the high levels of X5P and R5P (Text for Figure 1D) in WT strain are striking here which was mentioned in the text. It might be helpful to include the pooled results of measurements of R5P and Ru5P and add it to Figure 1D.

We added the levels of the combined metabolites to Fig 1D.

you could include an additional figure in supplement which gives more transparency of all measured products of X5P and R5P -> to emphasize that all products show lower amounts in WT cells (not only adenosine)

Besides adenosine, the manuscript also shows measures of the levels of the R5P products DNA, RNA and FAD.

The metabolite X5P seems accumulated in WT cells -> does X5P have other roles, for example in signaling? Could there be the option that its accumulation might display an advantage for antiviral response? -> it is for example known that Xu5P is a signaling compound and involved in activation of PPases controlling fat synthesis and glucose metabolism (PMID: 7929321, PMID: 12684532)

This is an appealing idea but currently isn't supported. Subsequent reports claim that G6P, rather than X5P, controls glucose metabolism via ChREBP activity (PMID: 21835137).

Did you ever test also macrophages derived from liver cells beside the tested spleen and bone-marrow derived macrophages? The ribokinase activity seems for example to be tissue specific, so would be interesting to see if the mechanism is still similar in this liver tissue

We haven't tried this but we agree with the notion. Perhaps not macrophages from different tissues (which we expect to respond similarly) but the cells that form different tissues. Besides the liver's capacity to use ribose, kidney cells appear to have an enhanced capacity to use fructose. Both tissues are reservoirs for infectious viruses (e.g. Polyomaviridae and Hepatitis viruses). This has been proposed to be a consequence of immune privilege but our findings raise the concept that this could also be due to alternative carbon metabolism that may advantage virus replication.

Figure 3: suggestion to use different colors as it's a bit confusing to have the same colors orange and blue in a slightly different context (Figure 3B)

The plots have been edited to use different colours in these different contexts to avoid confusion.

Referee #1:

Although WT and PKR knockout cells are used in most experiments, there is no prior characterization of their biochemical and immunological status towards Poly:I:C exposure documented. This includes the state of eIF2A phosphorylation, protein synthesis levels, and cytokine production, including type-I IFN. The absence of evidence regarding a decrease in eIF2A-P levels and the potential lack of translational arrest in PKR-KO cells raises questions about how these may result into [sic] the observed metabolic changes in Figure 1.

The cells used are isolated from a transgenic mouse that is well-characterised (PMID 8557029) and we have previously reported on the phenotype of these specific cells (e.g. PMID 26794869 & 18625702). The immunoblot of PKR confirmed the cell response to pIC treatment by the induction of the IFN response (Appendix Fig SF2D). Micrographs of the cell phenotype and the induction of ATF3 upon treatment with pIC have been added (Appendix Fig S2A & B).

The claimed absence of evidence is cryptic. The evidence for the involvement of PKR in the response is inherent in the experiments using cells with a specific gene mutation. This is repeated by knockdown of the transcript to identify the dependence for PKR expression. The dependence on kinase activity is also confirmed by comparison of active and point mutant, kinase-dead molecules in mammalian cells and by expressing specific constructs in bacteria. Importantly, the experiments in bacteria unequivocally exclude the established activities of PKR in translation control. Effects from ancillary pathways are also excluded by in vitro experiments with purified proteins.

Higher glucose consumption due to increased protein synthesis could significantly modify metabolism.

The levels of most metabolites in the PPP are lower in the WT cells compared to the PKR-ablated cells, i.e. opposing the proposition of PKR-dependent suppression of energy-consuming translation.

Experiments should have been conducted in WT cells using ISRIB and/or a PKR inhibitor to confirm or refute the results obtained with PKR KO cells. Importantly, inhibiting protein synthesis using an EIF4G1 inhibitor could mimic EIF2AK activation in PKR KO cells and confirm or refute the direct role of PKR in interfering with the PPP.

The tests used genetic knockout and knockdown and a kinase-dead molecule to directly test PKR activity. These tests are more specific than using small molecule binders of the GTP exchange factor eIF2B, recruitment of mRNA to the ribosome via eIF4G1 or even PKR inhibitors, which all suffer from off-target effects. However, the effects of three inhibitors (Salubrinal, ISRIB and 2-aminopurine) on the expression of TKT are shown in the supplementary data (Appendix FS2E).

Additionally, different autocrine responses to IFN- β in WT or EIF2KA cells upon Poly-I:C delivery could also impact glucose metabolism. Experiments in IFNAR KO cells or in the absence of effective IFN- β activity should be added.

PKR does not have a strong autocrine effect and the PKR ablated cells still respond to double-stranded RNA (via Rig1, Tlr, Ifit1, Oas, etc) so these responses are captured in the experiments. As stated, biochemical experiments also directly confirm the activity of PKR in vitro free of other cell signalling pathways and the experiments in bacteria negate any difference in the proposed autocrine responses as they are absent.

Knocking out IFNAR severely impacts cytokine signalling and significantly alters the expression of IFN-regulated genes, many of which control the immune response to RNA. As a result, this experiment suffers from the complications that the Reviewer raises.

The method and delivery of poly I:C are poorly described.

The methods section described that the pIC was delivered to the macrophages by complexing with the lipofectamine reagent. This description is now repeated in the results section.

It's important to consider that parallel activation of TLR3 could trigger IFN in macrophages independently of intracellular delivery, potentially altering cell reactivity.

TLR3 signalling is independent of PKR and is retained in the PKR-ablated cells so it doesn't account for the different responses of the WT and KO or KD cells.

The use of a prokaryotic assay to monitor eukaryotic kinase activity through overexpression is problematic. Yeast and yeast two-hybrid systems might have been more suitable.

No justification for this opinion is given but may refer to the novelty of the bacterial experiments while previous reports have explored PKR activity in *S. cerevisiae*. The experiments are conceptually similar. Just as the conservation of the yeast Sui2 to human EIF2 α makes it a PKR substrate, the conservation of the *E. coli* isomerase makes it a substrate for human PKR. Importantly, *E. coli* has the utility to be able to metabolise different single-carbon sources via the PPP to allow dissection of the PPP. This resourcefulness is not present in mammals or even yeast. Importantly, *S. cerevisiae* can uniquely generate ribose 5-phosphate by riboneogenesis, thereby reducing the role of the PPP. We include analysis to demonstrate the conservation of the isomerase by its amino acid sequence (F6D), and tertiary structure (Appendix FS5) and by assessing its interactions with the human homolog and the kinase (Appendix FS9).

The yeast-two-hybrid system is a method to assess protein-protein interactions and so may be mistakenly mentioned.

The phosphatase used to inhibit PKR activity in *E. coli* is not identified.

The material section and figure legend identified this as the bacteriophage Lambda phosphatase. We repeated this in the results section.

The experiments lack essential controls, including the expression of other eIF2A kinases besides PKR and/or kinase domains, to demonstrate the specificity of the observation and at least rule out off-target effects.

The specificity of the targeting is shown in a variety of experiments that are all controlled, including gene knockout and knockdown, over-expression of proteins and comparison of the WT and kinase-inactive PKR molecule, as well as in vitro experiments with purified proteins and experiments in bacterial using specific proteins and single carbon sources that are independent of the EIF2 α pathway.

It isn't clear what off-target effects are envisaged as PKR doesn't control the expression of the other EIF2 α kinases. The proposition may be that ablating PKR could heighten translation, which could activate GCN2 or PERK by reducing the pool of amino acids or promoting the accumulation of unfolded proteins. However, such redundant activity by the other kinases would obscure any effect of ablating PKR. The alternative possibility, that the effects are entirely a consequence of the heightened activity of other EIF2 α kinases in the absence of PKR, is excluded by experiments that do not include the other kinases.

Conservation of the EIF2 α kinases could mean that they may also phosphorylate RPIA and so are not a clear negative control. The primordial member of this kinase family, GCN2, is also controlled by nucleic acids (tRNA) so there is the opportunity for positive selection to install the development of the biological response, possibly before the development of the other kinases.

In Figure 4, contrary to the text description, protein levels in the blot appear different in WT and [sic] eIF2K -/- extracts for G6PD, PDGLS, PDG, RPIA, TKT, and GDPDH. These experiments need to be repeated and quantified more accurately to support the authors' claims. This again relates to the status of protein synthesis in the cells (not documented) and its overall effect on metabolism.

The immunoblots have been replaced with a graph showing the level of proteins measured in repeated experiments as had previously been described in the text. The immunoblots have been moved to supplementary data.

Relatedly, we revised the plot of the levels of the gene transcripts (Fig 3B, previously 2A). Formerly we referenced the levels of the mRNA against *Gapdh*. However, examination of gene array data and immunoblots showed that the levels of this factor

were inconsistent. This experiment used two other controls so we replotted the data referencing one of these. As a result of this correction, we now record a modest decrease of *Pgls* and *Rpe* and an increase in *Tkt* in the Null cells compared to the WT cells treated with pIC (p=0.0432, =0.0268 & =0.037, respectively). The text has been modified accordingly, although there is no change in the interpretation of the findings. We also reintroduced the WT to the plot (Fig 3B). This had been excluded it was used as the control to normalise the data and so is invariant, but on reflection, this may have been confusing. We also now included the measures of transcripts before normalisation (Appendix FS2C).

The efficiency of silencing by shRNA is not documented.

The efficiency of PKR knockdown in macrophages was shown (Appendix FS2F) and is now stated in the text.

The rationale for changing cell types and gene inactivation technique at this stage of the manuscript is unclear.

We now present this data earlier (data in Fig 4 moved to Fig 2 & 3). The rationale for the experiment is to ablate the kinase through a different method, which might differently impact the cells. The duplicated response ensures that the observed observations are not due to an unanticipated effect. Also, the comparison of the same cell expressing different shRNAs might be considered a closer comparison than cells isolated from separate (isogenic) animals.

shRNA can modify the IFN response by triggering innate receptors and could have extremely different impact than gene deletion.

A nontargeting shRNA controls for these effects. The KD cells replicated the response of the KO macrophages.

Again the impact on eIF2A phosphorylation, protein synthesis levels and cytokine production during silencing is not documented.

The gene knockout and knockdown experiments capture cytokine responses to RNA so this can't explain the response -i.e. PKR ablated cells still respond to RNA through other nucleic acid-binding proteins. Most conclusively, the effect of PKR on the metabolism is demonstrated in a system that is independent of the translation controlled by PKR so these effects are ruled out.

Comparison of cell lines used in different assays by RNAseq would provide a clearer view of the differences induced by PKR deficiency and support the authors' conclusions.

The effect of PKR activation in the time frame of the experiments is principally translational so is best captured through the direct measures of protein levels, their

activity and posttranslational state as well as the effect on metabolites, as performed in our manuscript. However, we also measured the levels of the PPP transcripts.

We have previously reported transcriptome analysis of the WT and *Eif2ak2*^{-/-} spleen macrophages by microarray. Notably, a primary response to PKR activity is the induction of the EIF2alpha phosphatase *Ppp1r15b*. However, this is not evident by analysis of transcripts, in keeping with its expression being induced translationally by derepressing its suppressive upstream open reading frame. This microarray data shows levels of the *G6pd*, *RpiA*, *Tkt* and *Taldo1* transcripts are equivalent and distinguished no inherent difference in RNA receptors such as *Rig1*, or the transcripts of associated signalling proteins such as *Irf3* and *Stat1* between the WT and *Eif2ak2*^{-/-} cells treated with pIC for 1, 3 and 6 hrs.

In Figure 4L, the authors use several useful controls, but they are [not] used in the PKR KO line, and are introduced late in the paper, without any demonstration of successful ectopic expression shown.

All treatments were also tested in PKR KD cells. Where there was no effect the results weren't shown, except for the expression of the rate-limiting enzyme of the oxidative phase of the PPP as a representative (Fig 2G&H). Treatments that showed a difference are shown. The constructs are expressed as a cleavable polyprotein with the blasticidin resistance gene under continuous selection.

ATF4 silencing/downregulation (again not documented) seems capable of counteracting PKR inactivation, suggesting the existence of a relationship established between the PPP cycle and the ISR. However, the authors quickly disregard this observation, that nevertheless indicates a potential role for translation control (ATF4 is normally translated upon protein synthesis arrest) in controlling glucose metabolism, despite the lack of documentation of ATF4 expression and translation in their experimental setting.

ATF4 expression isn't solely dependent on PKR and the effect of ATF4 on FAD was not changed in the PKR-ablated cells. Accordingly, we discounted this as a major contributor to the PKR-dependent control of FAD levels. We include the data of coordinately suppressing ATF4 with PKR to support this explanation (now Fig 3A).

In Figure 5, the experiments should also be performed with a specific PKR inhibitor to exclude any contaminating kinase activity coming from the purification procedure.

The in vitro kinase assay was conducted with and without PKR to exclude activity from kinases copurified with the PPP enzymes. Autoradiographs of assays without PKR are blank and so are unenlightening. We include autoradiographs to record the activity of the purified recombinant PKR protein and to demonstrate there is no phosphorylation in the RPIA protein preparation in the absence of the kinase (Appendix FSF7).

It is important to clarify how PKR is activated in the assay.

The methods section stated that the kinase was activated with pIC and cites the original protocol.

Additionally, controlling the specificity of the activity with another eIF2aK should be done.

Our use of the kinase-dead PKR molecule that is used in many of the experiments is a better control for PKR's phosphorylation activity.

In Figure 6, the quality of the microscopy is poor, lacking quantification and secondary labeling for G3BP1-positive stress granules known to contain PKR upon eIF2 phosphorylation. The figure does not adequately support the conclusions in the text.

We uploaded a higher-resolution figure and changed one of the panels to improve the quality of the figure.

The purpose of this quantification isn't clear as the objective is to show the spatial pattern of the proteins rather than making a statement about their relative expression. This is merely discussed to raise the contingency that kinase activity may affect the protein interaction.

The Venus signal doesn't accord with that expected for stress granules and biochemical analysis of stress granules does not support the enrichment of RPIA in these structures (PMID 36662637). We included an image that demonstrated that exogenous PKR is expressed cytosolic-wide and is not restricted to stress granules. This has previously been reported by ourselves (e.g. PMIDs 22633459 & 25404612) and others (PMID 33984068). We include some additional images of the Venus signal to clarify the function and expression pattern of Venus-tagged proteins (Appendix FS8&9).

The effect of the JNB inhibitor on HSV1 replication is perplexing, showing a block on the viral GFP reporter and an effect on the relative replication titer. A careful study of the inhibitor's impact should be performed in a separate manuscript, independently of the PKR story, as PPP modulation is currently proposed as an anti-viral target.

The cause for the confusion isn't clear but may refer to the difference between the level of GFP compared to the virus titre that was discussed above as a consequence of different controls of translation and nucleic acid synthesis.

The authors have not directly demonstrated that phosphorylation of RPIA by PKR has an effect on viral expression.

Agreed. We have not been able to eliminate endogenous ribose 5-phosphate production to be able to test the impact of a phosphor mutant isomerase for virus replication. We have established the rest of the response, showing PKR controls metabolism via the PPP and, more specifically, controls RPIA activity, demonstrating that PKR's kinase activity is required and that RPIA is directly phosphorylated by PKR. Accordingly, we contend that there is sufficient support for the proposition.

Additionally, the conclusion on the effect of RPIA phosphorylation (or lack thereof) by PKR in *E. coli* is confusing, suggesting again an off-target effect of the kinase on bacterial growth.

RPIA is demonstrated to be phosphorylated by PKR in *E. coli* (Fig 5). The PKR-dependent repression of bacterial growth is a direct consequence of this phosphorylation. This experiment is analogous to the yeast experiment that was referenced by the Reviewer, whereby the similarity of the *E. coli* isomerase to RPIA makes it a substrate for PKR.

Minor: Figure numbers should be indicated on the figures, and the manuscript's writing could be considerably simplified.

Apologies, for the absence of figure numbers.

We have tried to simplify the writing (as described above).

Dear Dr Sadler,

Thank you for submitting your revised manuscript (EMBOJ-2023-115586R) to The EMBO Journal. Your amended study was sent back to the three referees for their scientific re-evaluation, and we have received detailed comments from all of them, which I enclose below.

As you will see, the experts state that the work has been improved by the revisions and referees #2 and #3 are now in favour of publication, pending minor revision. However referee #1 points to remaining important caveats and ambiguities which in his/her view decrease the relevance of the work. In more detail, this referee requests clarification of the usage of dsRNA throughout the manuscript (ref#1, pt.1), as well as appropriate control experiments to disentangle PKR's function on PPP from its reported ISR function (ref#1, pt.2) and type I IFN stimulation of and production downstream of PKR (ref#1, pt.3).

These are relevant points in our view, in we concur it is important to put your novel findings into appropriate context.

We thus invite you here to revise the manuscript in a final revision, considering the remaining comments of referee #2 carefully and amending the manuscript accordingly.

Also, we now need you to take care of a number of issues related to formatting and data presentation as detailed below, which should be addressed at re-submission.

Please contact me at any time if you have additional questions related to below points.

Thank you for giving us the chance to consider your manuscript for The EMBO Journal. I look forward to your final revision.

Again, please contact me at any time if you need any help or have further questions.

Kind regards,

Daniel Klimmeck

>> Please add up to five keywords to your manuscript.

>> Author Contributions: Please remove the author contributions information from the manuscript text. Note that CRediT has replaced the traditional author contributions section as of now because it offers a systematic machine-readable author contributions format that allows for more effective research assessment. and use the free text boxes beneath each contributing author's name to add specific details on the author's contribution.

More information is available in our guide to authors.
<https://www.embopress.org/page/journal/14602075/authorguide>

>> Adjust the title of the 'Conflict of Interests' section to 'Disclosure and Competing Interests Statement'.

>> Funding information should be included in Acknowledgements.

>> Appendix: add page numbers to the ToC on the first page; Schemes S1 and S2 should be renamed to Appendix Figure S1-S2 with the corresponding callouts and included in ToC; all the other figures should be renamed to be in the consecutive order - Appendix Figure S3-S11 with updated callouts.

>> Source data files need to be reorganized to one file/folder per figure and ZIPing for each main figure.

>> Please indicate redisplay of data from Figure 3F in the figure legend for Appendix Figure S3.

>> Reference format: please adjust to EMBO Journal format, alphabetical, 10 authors + et al. .

>> Data Availability Section: please remove referee token and ensure privacy is released.

>> Author checklist: provide complete the 'Experimental animals' section.

>> "Supplementary Materials" section should be removed from the manuscript file.

>> Consider additional changes and comments from our production team as indicated below:

Figure Legends (main + EV): "1. Please note that a separate 'Data Information' section is required in the legends of figures 2a-j; 3a-d, g; 4a-b, d; 7b-e; 8f-h.

2. Please note that the legend for figure 3c is incorrectly labelled as 3b in the manuscript. This needs to be rectified."

Please indicate the statistical test used for data analysis in the legends of figures 8c-e.

"1. Please note that the box plot needs to be defined in terms of minima, maxima, centre, bounds of box and whiskers, and percentile in the legend of figure 3g.

2. Please note that information related to n is missing in the legend of figure 2b.

3. Although 'n' is provided, please describe the nature of entity for 'n' in the legends of figures 2a, c-j; 3a-d, g; 4a-b, d; 7b, e; 8c-h.

4. Please note that the error bars are not defined in the legends of figures 8c-e."

Please note that the white arrows are not defined in the legend of figure 5e. This needs to be rectified.

Referee #1:

In this revised version, the authors have addressed some concerns raised in previous reviews. However, it remains unclear to me whether the Poly I:C stimulation, utilized in the initial experiments illustrated in Fig. 1, is also applied in other experiments detailed in the manuscript.

The results in Figure 1, as described in line 15 of the results section, reveal that glucose metabolism is affected during Poly I:C activation of PKR. This sets the stage for later demonstrations in the paper regarding the potential antiviral properties of inhibiting the PPP.

1) Surprisingly, there is no mention or description of dsRNA stimulation in the subsequent experiments investigating the mechanisms impacting the PPP. Is this omission accurate, or is it absent from the descriptions and figure legends?

If dsRNA stimulation was not employed, it is perplexing. Without stimulation, PKR activity is presumably relatively low at a steady state, making it challenging to explain how the kinase contributes to glucose metabolism regulation. Establishing the level of PKR activation in different cells at steady state or upon stimulation should provide clarity in that matter and potentially validate several of the control experiments presented in the paper (ISRIB, Salubrinal, expression of eIF2B etc..). In particular comparing steady state and Poly I:C treated cells, for the experiments depicted in Fig. 2, should be a clear demonstration of the role of PKR in regulating glucose metabolism, at steady state or upon infection-like conditions.

2) Importantly, the authors deemed measuring protein synthesis and eIF2a phosphorylation, or altering protein synthesis (eIF4G inhibition) in WT and KO cells was irrelevant to their purpose. However, if PKR activity genuinely regulates metabolism at steady state, it is crucial to document and demonstrate that its ability to phosphorylate RPIA is independent of dsRNA activation and its

role in the ISR.

3) Alternatively, if dsRNA stimulation is necessary to alter glucose metabolism, the role of type-I IFN in this process must be documented. Both PKR and PPP1R15A are ISGs, and PKR KO cells exhibit a different capacity to produce IFN (at least at the protein level) than their WT counterparts in response to dsRNA, contrary to what is stated in the rebuttal letter.

Referee #2:

The authors have addressed the Reviewer's concerns in the point-by-point letter, I feel however they have done about the minimum, and as a result, the revised paper is better as the first version, but equally I feel it could have become better given the input provided. I don't object the publication, but equally, I feel the manuscript could have evolved more to make a stronger case.

Referee #3:

This manuscript is substantially improved and makes the significant contribution that PKR phosphorylates RPIA upon dsRNA sensing and this can contribute to the antiviral function of PKR. I recommend publication.

However, before publication a few minor issues should be fixed by the authors as detailed below:

1) page 15, line 4. Please be explicit in the text about whether this metabolic analysis is done with or without dsRNA transfection (I believe not?). This is important since if just done with normal cells it demonstrates that there is sufficient PKR activity (perhaps due to endogenous dsRNAs?) to regulate the PPP.

1b) I would also recommend a short description of the PPP here since many readers of this work will not be familiar with that pathway.

2) page 16, line 25. This sentence is grammatically incorrect and should be edited.

3) page 17, line 5. Again, I recommend being explicit about whether there is added dsRNA or not in the text. This will help the readers follow the analyses.

4) page 17, line 30. I recommend stating changing "didn't alter cell fluorescence" to "didn't alter the levels of FAD, as assessed by intrinsic cell fluorescence.

5) Figure 3A. I recommend including the sh-P and shP+WTPKR sample in this histogram to allow more complete presentation of the critical data. (Is the K296R PKR gene shRNA resistant? If so, please state in the text).

6) Figure 8H. The difference between the two graphs just doesn't make sense. How can the 051A and IFN β samples be roughly the same in the left panel, and then dramatically different in the right panel after standardization to the same control. This should be fixed.

Referee #1:

In this revised version, the authors have addressed some concerns raised in previous reviews. However, it remains unclear to me whether the Poly I:C stimulation, utilized in the initial experiments illustrated in Fig. 1, is also applied in other experiments detailed in the manuscript.

The results in Figure 1, as described in line 15 of the results section, reveal that glucose metabolism is affected during Poly I:C activation of PKR. This sets the stage for later demonstrations in the paper regarding the potential antiviral properties of inhibiting the PPP.

1) Surprisingly, there is no mention or description of dsRNA stimulation in the subsequent experiments investigating the mechanisms impacting the PPP. Is this omission accurate, or is it absent from the descriptions and figure legends?

RESPONSE

The spleen macrophages were treated with pIC in all experiments. This had been described in the material and methods section and at the beginning of the results section. It is now repeated on each occasion this occurs under the heading; *Assays in mammalian cells* (p7, ln 20-24), *Isoelectric focusing* (p9, ln20), *PKR alters glucose metabolism* (p15, ln 2-3), *PKR limits ribose 5-phosphate production* (p16, ln28) and later under the heading *Phosphor-control of the Ribose 5-phosphate isomerase* (p22, ln28).

This information is not repeated in the legends of the main figures, as the text captures it. However, this is repeated in the legends of the figures in the appendix (S4, S9 & S10).

In difference to the splenic macrophages, no pIC stimulation was used in experiments in transformed/transfected cells as the exogenous vectors themselves produce duplex RNA. This explanation has been noted in the M&M section (p7, ln22-24) and citations supporting this have been added (PMID 30838421 & 24475301).

1...) If dsRNA stimulation was not employed, it is perplexing. Without stimulation, PKR activity is presumably relatively low at a steady state, making it challenging to explain how the kinase contributes to glucose metabolism regulation. Establishing the level of PKR activation in different cells at steady state or upon stimulation should provide clarity in that matter and potentially validate several of the control experiments presented in the paper (ISRIB, Salubrinal, expression of eIF2B etc..). In particular comparing steady state and Poly I:C treated cells, for the experiments depicted in Fig. 2, should be a clear demonstration of the role of PKR in regulating glucose metabolism, at steady state or upon infection-like conditions.

RESPONSE

The experiments include activating stimuli so the speculation of PKR function without activation is unwarranted.

Activating stimuli are essential for PKR function. The kinase is held in an auto-inhibited conformation in which an association with the kinase's substrates is blocked by an interaction with one of the two amino-terminal RNA-binding domains and the dimerisation of PKR, required for autocatalysis, is suppressed by the second RNA-binding domain's association at the dimer interface. RNA disrupts these inactivating intramolecular associations by engaging the RNA-binding domains. Publications over more than forty years have established this, so it doesn't need to be re-examined (e.g. PMID 6179946, 3479429, 33911136, 28281686, 11032824, 17011579, 16179258, 16179259).

As an aside, in vitro culture of cells and immortalisation activates PKR to some extent. Culture conditions create duplex RNA through the aberrant control of transcripts and base modification and the release of nucleic acids from mitochondria. Accordingly, cultured cells fluctuate between a lesser/greater degree of PKR activation as opposed to the envisaged inactive states. Our experiments comparing the WT and PKR-ablated cells capture PKR function irrespective of the extent of the molecule's activity.

2) Importantly, the authors deemed measuring protein synthesis and eIF2a phosphorylation, or altering protein synthesis (eIF4G inhibition) in WT and KO cells was irrelevant to their purpose. However, if PKR activity genuinely regulates metabolism at steady state, it is crucial to document and demonstrate that its ability to phosphorylate RPIA is independent of dsRNA activation and its role in the ISR.

RESPONSE

Our dismissal of the need to measure the extent of PKR's phosphorylation of EIF2S1 or the ensuing effect on EIF4G activity derived from our demonstration that PKR controls the PPP in contexts that are entirely independent of these factors.

We do not suggest that PKR functions without RNA activation so do not believe it is necessary to test this notional mechanism.

3) Alternatively, if dsRNA stimulation is necessary to alter glucose metabolism, the role of type-I IFN in this process must be documented. Both PKR and PPP1R15A are ISGs, and PKR KO cells exhibit a different capacity to produce IFN (at least at the protein level) than their WT counterparts in response to dsRNA, contrary to what is stated in the rebuttal letter.

RESPONSE

We refuted the need to assess the type I and III interferon responses because PKR was shown to control the PPP in a system that doesn't produce these cytokines and doesn't have the cell signalling pathway.

We also noted that the interferon response is retained in the mammalian cell experiments comparing WT and PKR ablated cells and so was unlikely to account for the differences. We concede that this second point is less equivocal, although more potent inducers remain in the absence of PKR the kinase does induce the cytokines. However, PKR also functions in cell signalling as a bridging molecule, without substrate phosphorylation, so the kinase-dead molecule retains the capacity to induce interferons (PMID 10848580, 16600570, 15121867, 14749731, 35522180) but the expression of this kinase-dead molecule (K296R) didn't restore the metabolism of PKR-ablated cells.

As is also suggested, PKR could repress the expression of interferon-induced genes. However, this largely rescinds PKR's induction of interferons so is of dubious impact. The greater effect of interferons is paracrine signalling to prime protective functions in uninfected cells with inactive PKR, rather than autocrine signalling in cells with already active PKR. This shifts later when many more cells become infected. However, the sustained PKR activity (if not blocked by viruses) at this point would induce cell death, making the effects on the metabolism less significant.

Regarding the induction of the genes by interferon, examining the effect of the IFN response on the induction of PKR is rendered meaningless in experiments that compared WT and PKR ablated cells. Also, there isn't strong support for the contention that PP1R15A is an interferon-stimulated gene. The transcript isn't induced in our experiments and computational analysis (MotifMap, ENCODE or CHEA) doesn't identify the expected promoter motifs, CHIP databases (ENCODE & CHEA) don't identify the expected transcription factors and perturbation of transcription factors and kinases involved in the IFN response don't alter its expression (GEO Signatures, KnockTF Gene Expression Profiles & LINCS L1000 CMAP Signatures). More applicable to our report, this analysis also applies to the PPP enzymes, and we confirmed that their expression isn't altered.

The functions of PKR in the control of translation through EIF2S1 phosphorylation and in cell signalling with cytokine induction undoubtedly occur and will affect cell function but are not the mechanism by which PKR controls the PPP so we do not re-examine these established activities but focus on the new activity of PKR we have identified.

Referee #2:

The authors have addressed the Reviewer's concerns in the point-by-point letter, I feel however they have done about the minimum, and as a result, the revised paper is better as the first version, but equally I feel it could have become better given the input provided. I don't object the publication, but equally, I feel the manuscript could have evolved more to make a stronger case.

NO RESPONSE

Referee #3:

This manuscript is substantially improved and makes the significant contribution that PKR phosphorylates RPIA upon dsRNA sensing and this can contribute to the antiviral function of PKR. I recommend publication.

However, before publication a few minor issues should be fixed by the authors as detailed below:

1a) page 15, line 4. Please be explicit in the text about whether this metabolic analysis is done with or without dsRNA transfection (I believe not?). This is important since if just done with normal cells it demonstrates that there is sufficient PKR activity (perhaps due to endogenous dsRNAs?) to regulate the PPP.

RESPONSE We have clarified our use of RNA stimuli throughout the manuscript as described in our response to similar comments by Reviewer 1.

1b) I would also recommend a short description of the PPP here since many readers of this work will not be familiar with that pathway.

RESPONSE

The following text was added to the beginning of the results section, under the heading *PKR alters glucose metabolism* (p15, ln19-29):

The PPP consists of two decoupled phases. An aerobic phase oxidises glucose 6-phosphate to produce carbon dioxide, NADPH and ribulose 5-phosphate. Subsequent isomerisation and epimerisation in a nonoxidative phase convert the ribulose 5-phosphate with fructose 6-phosphate and glyceraldehyde 3-phosphate from the glycolytic pathway into different intermediates, the most important of which is ribose 5-phosphate and returns unused carbon to the glycolytic/gluconeogenic pathway.

2)page 16, line 25. This sentence is grammatically incorrect and should be edited.

RESPONSE

'rederived' replaces the incorrect 'rederivation'

3) page 17, line 5. Again, I recommend being explicit about whether there is added dsRNA or not in the text. This will help the readers follow the analyses.

RESPONSE

The use of pIC in this instance has been added.

4) page 17, line 30. I recommend stating changing "didn't alter cell fluorescence" to "didn't alter the levels of FAD, as assessed by intrinsic cell fluorescence.

RESPONSE changed as suggested.

5) Figure 3A. I recommend including the sh-P and shP+WTPKR sample in this histogram to allow more complete presentation of the critical data. (Is the K296R PKR gene shRNA resistant? If so, please state in the text).

RESPONSE

The data for the shP sample has been added to Fig 3A.

The shRNAs targeted the *Eif2ak2* UTRs outside of the gene's ORF that was used so the expression construct escapes the control. This explanation has been added to the text in the M&M under the heading *Assays in mammalian cells* (p7, ln 10-12).

As might be anticipated from our findings, re-expression of the WT PKR wasn't possible. Although puromycin-resistant colonies (selecting the transgene) were isolated they did not grow. Clones with shRNAs targeting RPIA behaved similarly. Interestingly, shRNA lines targeting TKT were isolated and appeared normal under passage but were sensitive to stress.

6) Figure 8H. The difference between the two graphs just doesn't make sense. How can the 051A and IFN β samples be roughly the same in the left panel, and then dramatically different in the right panel after standardization to the same control. This should be fixed.

RESPONSE

The data are treated differently in each graph. One approach averages logarithms and the other creates ratios between datasets. Averaging logarithms is effective for handling widely varying data but compresses the magnitude of differences between data points. This is better captured by ratios. The data displayed in Fig 8G is the average of the total viral titres in the five experiments plotted on the logarithmic scale. The data displayed in Fig 8H is standardised to the control in each experiment and then these ratios are averaged and plotted on a linear scale. This later standardisation prevents the swamping of the difference between treatments by variance between experiments. Virus production varied up to 1000-fold between the separate experiments while variance between treatments varied up to 100-fold within an experiment. This latter difference is obscured in the logarithmic data (Fig. 8G) but by converting the data to a ratio of virus titre in the treatment v the control (Fig 8H) it preserves the original information in the sense that it compares the values directly, with each ratio representing a comparison between the corresponding points in the dataset. The Reviewer's expectation (of equivalence between treatments) would be borne out if we were to convert the average of the logarithmic data to a ratio. However, this would be a superficial change in the presentation of the logarithmic data that maintained its compressive effects.

We include the following justification for our methodology in the results section under the title *Pharmaceutical targeting of PPP as an antiviral therapy* (p28, ln6-9) to support the description of the data in the legend:

(in text) This data is shown as an average of the total viral titre plotted on a logarithmic scale and, as this compresses the magnitude of differences between data points, also as a direct comparison of the treatments to the control in each experiment by conversion to a ratio that is plotted on a linear scale.

(in figure legend) The total virus titre is shown on the left as an average of all experiments plotted on a logarithmic scale and then replotted on the right as a ratio of the virus titre from each treatment compared to the control in separate experiments that are then averaged and shown on a linear y-axis

Dear Dr Sadler,

Thank you for submitting the revised version of your manuscript. I have now evaluated your amended manuscript and also received final input from referee #1 on your revisions. In light of all information at hand, we concluded that the remaining minor concerns have been addressed to a sufficient level.

I am thus pleased to inform you that your manuscript has been accepted for publication in the EMBO Journal.

On a different note, I would like to alert you that EMBO Press offers a format for a video-synopsis of work published with us, which essentially is a short, author-generated film explaining the core findings in hand drawings, and, as we believe, can be very useful to increase visibility of the work. Please see the following link for representative examples and their integration into the article web page:

<https://www.embopress.org/doi/full/10.15252/emboj.2019103932>

Best regards,

Daniel Klimmeck

Daniel Klimmeck, PhD
Senior Editor
The EMBO Journal
EMBO
Postfach 1022-40
Meyerhofstrasse 1
D-69117 Heidelberg
contact@embojournal.org
Submit at: <http://emboj.msubmit.net>

Referee #1:

Although the authors dismissed a bit quickly my suggestions and did not really address them experimentally, I am supporting publication.
